# Overexpression of *Tamarix hispida* ThTrx5 Confers Salt Tolerance to Arabidopsis by Activating Stress Response Signals

**DOI:** 10.3390/ijms21031165

**Published:** 2020-02-10

**Authors:** Jiayu Luan, Jingxiang Dong, Xin Song, Jing Jiang, Huiyu Li

**Affiliations:** State Key Laboratory of Tree Genetics and Breeding, Northeast Forestry University, Harbin 150040, China; jiayuluan@126.com (J.L.); dongjingxiang@126.com (J.D.); songxinaa@126.com (X.S.); jiangjing@nefu.edu.cn (J.J.)

**Keywords:** thioredoxin, salt stress, pull-down, transcription

## Abstract

Salt stress inhibits normal plant growth and development by disrupting cellular water absorption and metabolism. Therefore, understanding plant salt tolerance mechanisms should provide a theoretical basis for developing salt-resistant varieties. Here, we cloned *ThTrx5* from *Tamarix hispida*, a salt-resistant woody shrub, and generated *ThTrx5*-overexpressing transgenic *Arabidopsis thaliana* lines. Under NaCl stress, the germination rate of overexpressing *ThTrx5* lines was significantly increased relative to that of the nontransgenic line; under salt stress, superoxide dismutase (SOD), peroxidase (POD), catalase (CAT), and glutathione levels and root length and fresh weight values of transgenic ThTrx5 plants were significantly greater than corresponding values for wild-type plants. Moreover, with regard to the transcriptome, comparison of differential gene expression of transgenic versus nontransgenic lines at 0 h and 3 h of salt stress exposure revealed 500 and 194 differentially expressed genes (DEGs), respectively, that were mainly functionally linked to catalytic activity and binding process. Pull-down experiments showed that ThTrx bound 2-Cys peroxiredoxin BAS1-like protein that influences stress response-associated redox, hormone signal transduction, and transcription factor functions. Therefore, this work provides important insights into *ThTrx5* mechanisms that promote salt tolerance in plants.

## 1. Introduction

Thioredoxin (Trx) proteins, a class of conserved, thermostable, acidic small proteins [1] that are widely found in chloroplasts, yeast, animals, and plants [2,3,4], share the highly conserved active site core sequence Trp-Cys-Gly-Pro-Cys (WCXXC). Within this core sequence, two Cys residues confer unique biological properties to Trx proteins by reversibly engaging in disulfide linkage formation that releases two electrons and two protons that serve as redox pairs in various physiological reactions [5]. Based on amino acid homology, 20 Trx homologs encoded by the *Arabidopsis thaliana* (Hereinafter *Arabidopsis*) genome can be assigned to two Trx families, family I and family II. Upon structural analysis [6], family I proteins can be divided into six subclasses: h, f, m, o, x, and y, with Trx protein expression patterns differing among various organisms and cellular sites. For example, m, f, x, and y subclasses are found in chloroplasts, h in the cytoplasm, and o and x within mitochondria. Trx-o is a newly discovered subclass of family I that has been shown to control activities of several mitochondrial proteins through sulfur bond reduction, while the organization and function of the Trx-h subfamily appears to be particularly complex. Nevertheless, most research to date has focused on Trx protein functions of *A. thaliana* and poplar (*Populus* spp.) models of nonwoody and woody plants, respectively.

In plants, studies investigating Trx system functions have shown that Trx proteins are mainly associated with resistance to stress, such as drought, heat, and oxidation, through the regulation of stress resistance genes. Notably, Trx contains cysteine residues that can participate in redox reactions catalyzed by thioredoxin reductase (TrxR), an nicotinamide adenine dinucleotide phosphate (NADPH)-dependent enzyme [7], whereby Trx, TrxR, and NADPH work together as an important thioredoxin-based antioxidant system in plants. Trx and TrxR each contain two cysteine residues located in close proximity to one another within a conserved protein core sequence. These cysteine residues can form reversible disulfide bonds that are crucially important to Trx and TrxR functions as reductants. In fact, studies have shown that this thioredoxin-based system can effectively reduce intracellular glutathione disulfide (GSSG) [8] as one of its major regulatory roles for controlling signal transduction or for maintaining cytoplasmic proteins in reduced states [9]. Moreover, this system also plays a vital role in preventing oxidative damage to nucleic acids, proteins, and cell membranes to promote survival of organisms living in aerobic environments [10]. *Tamarix hispida*, a deciduous shrub and an important salt-tolerant plant that thrives in high-salt environments, can form a natural forest on saline-alkali soil [11] and is an ideal species for studying plant salt tolerance mechanisms. In previous studies, nine *ThTrx* genes were identified through transcriptome analysis of *T. hispida*, three of which belong to the m subclass and are localized in the chloroplast (*ThTrx5*, *ThTrx9*, *ThTrx10*); four genes of m subclasses located in chloroplasts were found in the study of Arabidopsis: *THIOREDOXIN M-TYPE 1* (AT1G03680.1), *THIOREDOXIN M-TYPE 2* (AT4G03520.1), *THIOREDOXIN M-TYPE 3* (AT2G15570), and *THIOREDOXIN M-TYPE 4* (T3G15360.1) [12]. The results are shown in Appendix A. All m subclass genes contain the highly conserved WCGPC domain, and *ThTrx5* has the highest identity to the *AtTRX-M4* (AT3G15360.1) gene in *Arabidopsis*. Importantly, *ThTrx5* has been shown to play a role in the salt stress response using real-time quantitative PCR (polymerase chain reaction)-based monitoring of *T. hispida* gene expression [13]. Here, the full-length sequence of the *T. hispida ThTrx5* gene was cloned into plant expression vector pGWB5 and the resulting plasmid was used to genetically transform salt-sensitive *Arabidopsis*. Salt tolerance effects conferred to transgenic *Arabidopsis* by overexpression of acquired *ThTrx5* were revealed by measurements of fresh weight and reduced glutathione content, while salt tolerance mechanisms were deduced from results of pull-down experiments and transcriptome analysis.

## 2. Results

### 2.1. Acquisition and Salt Resistance Analysis of ThTrx5-Overexpressing Transgenic Lines

Specific primers derived from the *ThTrx5* gene were designed based on *T. hispida* transcriptome data and were used to amplify DNA containing the predicted ThTrx5 Open Reading Frame (ORF) sequence (length 564 bp) via RT-PCR to form products confirmed by DNA sequencing. Next, the *ThTrx5* gene was cloned into the pGWB5 vector using Gateway methodology (Appendix A). 35S is the promoter used for the expression of the gene. The resulting plasmid, 35S::ThTrx5, was transferred into *Arabidopsis* via the floral dipping, yielding homozygous transgenic lines that were subsequently propagated to the T3 generation. After rescreening to identify correctly constructed lines, four of these lines were designated as OE1–OE4. OE1 and OE2 exhibited highest levels of *ThTrx5* gene expression and thus were selected for subsequent experiments (Appendix A).

Transgenic and wild-type *Arabidopsis* seeds were sown onto solid medium containing different concentrations of NaCl for use in preliminary analysis of salt tolerance through the monitoring of germination rate. The results showed that after exposure to 100 mM NaCl, the WT seed germination rate was 33% and the average germination rate of transgenic seeds (OE1, OE2) was 98%. When the NaCl concentration was increased to 150 mM, the germination rate of WT seeds was 13% and the average germination rate of transgenic seeds (OE1, OE2) was 93% (Figure 1). After transgenic and WT seedlings at 7 d post-germination were treated with 150 mM NaCl for 48 h, they were phenotypically compared, revealing that leaves of WT plants were obviously chlorotic and yellow, while most plants of OE1/OE2 lines grew normally (Figure 2A). Chlorophyll measurement results showed significantly reduced chlorophyll content levels for both WT and transgenic lines, but chlorophyll levels in transgenic lines were significantly higher than WT control levels after NaCl stress exposure. Moreover, root lengths and fresh weights of OE1 and OE2 plants were greater than respective values for WT plants (Figure 1). When SOD (superoxide dismutase), POD (peroxidase), CAT (catalase), and glutathione (GSSG) levels in transgenic and WT plants were measured at 0 h and 48 h of stress exposure, the results revealed that expression levels of enzymes and glutathione in transgenic line plants were significantly greater than respective levels in WT plants (Figure 2).

### 2.2. Screening of Interacting Proteins

#### Pull Down

A pull-down technique was used to reveal which proteins ThTrx5 interacted with during induction of salt tolerance. Using a prokaryotic expression vector, pET41a, ThTrx fusion protein containing the glutathione-S-transferase (GST) Tag, was constructed and expressed in *E. coli* (DE3), with fusion protein expression confirmed via Western blot analysis using antibody specific for GST (Figure 3A–C). After binding of the fusion protein to a nickel column, the total proteins of wild-type and ThTrx5 transgenic lines after salt stress were passed through the column and eluted after incubation, respectively. Different bands of WT and transgenic *Arabidopsis* protein were separated by SDS-PAGE, (Figure 3D,E). Subsequently, differentially expressed proteins were analyzed by mass spectrometry, and resulting data were compared to a protein database (SwissProt 57.15). Five proteins were identified as ThTrx5 interaction partners: CATALASE 3, GLYCERALDEHYDE-3-PHOSPHATE DEHYDROGENASE, PEPTIDYL-PROLYL CIS-TRANS ISOMERASE (CYP20-3), ATP SYNTHASE SUBUNIT BETA, and 2-CYS PEROXIREDOXIN BAS1-LIKE PROTEIN.

### 2.3. Transcriptome Analysis

#### 2.3.1. Changes in ThTrx5 Gene Expression in *Tamarix Hispida* under NaCl Stress

To determine when *ThTrx5* gene responded most significantly under salt stress, we studied by qRT-PCR *ThTrx5* and *α-Tubulin* transcript levels in *T. hispida* at 0, 3, 6, 12, 24, and 48 h of salt treatment. During exposure of plants to 150 mM NaCl treatment, *ThTrx5* gene expression trends were observed for *T. hispida* (Figure 4). The expression level of the *ThTrx5* gene reached its highest value at 3 h; with the prolongation of NaCl treatment time, the *ThTrx5* expression level decreased to its lowest at 12 h and 24 h (<2-fold) and then increased, and then the expression quantity increased to nearly two times than the value of 24h at 48 h.

Studies have shown that homologous genes have similar expression patterns in different species [14]. Therefore, we selected 3 h of NaCl exposure as the time point for subsequent transcriptome sequencing and pull-down analysis in *Arabidopsis*.

#### 2.3.2. Illumina Sequencing

To analyze regulatory mechanisms employed by *ThTrx5* during exposure to salt stress, four cDNA libraries were constructed after plants were grown for 7 days post-germination. Two libraries were made from control nontransgenic plants grown under nonstress conditions for 0 h (designated WT-0h) and plants exposed to 150 mM NaCl for 3 h (designated WT-3h). Two additional libraries were constructed as described above from transgenic plants not exposed to stress (designated OE-0h) and plants exposed to 150 mM NaCl for 3 h (designated OE-3h). Libraries were sequenced using the Illumina deep-sequencing platform. A total of 7,639,214, 7,499,090, 7,165,453, and 7,318,039 raw reads were generated by Illumina sequencing. After adapter sequence and low-quality sequences were removed, a final total of 7,478,939, 7,298,276, 7,071,661, and 7,309,844 clean reads remained, representing 97% of sequence obtained before sequence cleanup (Table 1).

#### 2.3.3. Screening and Analysis of Differentially Expressed Genes

At 0 h of NaCl stress exposure, 500 differential genes were detected in transgenic lines compared with the WT line, of which 115 genes were upregulated and 385 genes were downregulated. At 3 h of NaCl stress exposure, the number of differentially expressed genes in transgenic lines versus the WT line was 194, of which 66 genes were upregulated and 128 genes were downregulated. A total of 458 DEGs were specifically expressed in transgenic lines at 0 h of stress exposure, while 152 DEGs were specifically present in transgenic lines after 3 h of exposure (Figure 5).

#### 2.3.4. Analysis of GO (Gene Ontology) Enrichment of DEGs

Functional category enrichment analysis of DEGs between nontransgenic and transgenic lines exposed to NaCl stress was conducted based on the Gene Ontology (GO) database (*p*-value < 0.05, fold-change > 2). Significant GO terms were determined based on a cut-off value of *p* < 0.05. GO functional classification of differential genes in the WT 0 h vs. OE 0 h transcriptome revealed that DEGs annotation assignments fell into three categories: biological process, cellular component, and molecular function. For the biological process category, DEGs of transgenic lines were enriched for more genes belonging to subprocesses of metabolic process, cellular process, and single-organism process than were enriched in WT lines. In the category of molecular function, differential genes were mainly concentrated in catalytic activity and binding classes. In the category of cellular component, differential genes were mainly concentrated in cell, cell part, and organelle classes. GO analysis revealed differential expression of genes related to binding and catalytic activity in the transgenic lines, including *PEROXIDASE9* (AT1G44970.1), *CALMODULIN-LIKE 41* (AT3G50770.1), *Catalase 3* (AT1G20620.1), *CATION EXCHANGER 3* (AT3G51860.1), and *FE SUPEROXIDE DISMUTASE 1* (AT4G25100.2), with expression levels of these genes significantly upregulated, respectively, over corresponding wild-type levels. Enrichment of DEGs involved in these processes is likely linked to the transfer of the *ThTrx5* gene into *Arabidopsis*. Meanwhile, GO analysis of differential genes in WT 3h vs. OE 3h was also performed. The results showed that enriched DEGs fell within three major categories and were significantly enriched in biological process classes including metabolic process, cellular process, single-organism process, and response to stimulus. For the cellular component category, DEGs were mostly enriched in cell, cell part, and organelle classes. For the molecular function category, DEGs were mostly enriched in binding and catalytic activity classes. GO analysis revealed that DEGs related to binding and response to stimulus pathways were enriched in transgenic lines after salt stress and included *PEROXIDASE 52* (AT5G05340.1), *JASMONATE-REGULATED GENE 21*(AT3G55970.1), *DARK INDUCIBLE 11* (AT3G49620.1), and *BRUTUS* (AT3G18290), which were significantly downregulated by 4-fold, 8-fold, 16-fold, and 4-fold, respectively, relative to WT expression levels, while the mRNA expression level of *GLUTATHIONE S-TRANSFERASE F2* was significantly upregulated by 8-fold. Therefore, DEGs for these processes may play an important role in salt tolerance of transgenic *Arabidopsis* (Figure 6).

#### 2.3.5. Analysis of Transcription Factor (TF) Families Represented by DEGs Identified during Salt Stress Exposure

After all unigenes obtained here were compared with the Plant Transcription Factor Database (Plant TFDB), the three TF families with highest numbers of unigene matches included NAC, bHLH, and ORG TF families. Expression patterns of 14 TFs within 10 TF families were significantly different between transgenic and WT lines at 0 h, of which transcriptional expression of 4 TFs were upregulated, including *NAC DOMAIN CONTAINING PROTEIN 1* (AT1G56010.2), *NAC DOMAIN CONTAINING PROTEIN 102* (AT5G63790.1), *WRKY DNA-BINDING PROTEIN 70* (AT3G56400), and *YELLOW STRIPE LIKE 1* (AT4G24120.1); meanwhile, transcription levels of 10 TFs were downregulated, such as *AP2/B3-LIKE TRANSCRIPTIONAL FACTOR FAMILY PROTEIN* (AT1G01030), *MYB DOMAIN PROTEIN 48* (AT3G46130.1), *NAC DOMAIN CONTAINING PROTEIN 19* (AT1G52890.1), *SENESCENCE-ASSOCIATED GENE 21* (AT4G02380.1), and others. At 3 h of stress exposure, expression of 14 TFs within 10 TF families exhibited significant differences between transgenic versus nontransgenic lines, with 6 upregulated TFs identified that included *AP2* (AT4G34400.1), *MYB48* (AT3G46130.1), *NAC1* (AT1G56010.2), *SAG21* (AT4G02380.1), and others, while expression levels of 8 TFs were downregulated, such as *NAC domain containing protein 19* (AT1G52890.1), *NAC102* (AT5G63790.1), *BASIC HELIX-LOOP-HELIX TRANSCRIPTION FACTOR FAMILY PROTEIN101* (AT5G04150.1), *OBP3-RESPONSIVE GENE 3* (AT3G56980.1), among others (Figure 7). During salt stress exposure, upregulation of *AP2* expression was highest and was 3-fold higher than at 0 h, while downregulation of expression of *OBP3-RESPONSIVE GENE 2* (AT3G56970) and *ORG3* (AT3G56980.1) were most marked, with transcriptional expression levels exhibiting 2-fold and 3-fold reductions when compared with levels at 0 h, respectively.

#### 2.3.6. Analysis of DEGs Involved in Hormone Synthesis and Signal Transduction Pathways Triggered by Salt Stress Exposure

Analysis of DEGs involved in hormone signaling pathways during salt stress exposure revealed 13 genes associated with signal transduction pathways involving six hormones, namely ethylene (ETH), jasmonic acid (JA), 6-benzylaminopurine (6-BA), salicylic acid (SA), abscisic acid (ABA), and brassinosteroid (BR); these DEGs exhibited significant expression differences, with the greatest number of DEGs associated with ETH and JA pathways. At 0 h, *ERF DOMAIN 53* (AT2G20880.1), *GALACTURONOSYLTRANSFERASE-LIKE 2* (AT3G50760.1), *SQUALENE SYNTHASE 2* (AT4G34650.1), and *GLYCINE-RICH PROTEIN 5* (AT3G20470.1) were upregulated in transgenic *ThTrx5* versus WT lines, while nine downregulated genes included *ETHYLENE RESPONSE FACTOR-2* (AT5G47220.1), *ABI FIVE BINDING PROTEIN 1* (AT1G69260.1), *JASMONATE-REGULATED GENE 21* (AT3G55970.1), *VEGETATIVE STORAGE PROTEIN 1* (AT5G24780.1), and *EXTENSIN 3* (AT1G21310.1). At 3 h, three genes, including *ERF2* (AT5G47220.1), *EXT3* (AT1G21310.1), *GLYCINE-RICH PROTEIN 5* (AT3G20470.1), were induced in transgenic *ThTrx5* lines compared with the WT line, with 12 downregulated genes observed that included *ERF53* (AT2G20880.1), *AFP1* (AT1G69260.1), *JRG21* (AT3G55970.1), *VSP1* (AT5G24780.1), *VEGETATIVE STORAGE PROTEIN 2* (AT5G24770.1), and others (Table 2). Comparison of DEGs related to hormone signaling at 0 h and 3 h of stress exposure indicated that upregulated expression level of *ERF2* was highest and was 3-fold greater at 3 h than at 0 h, while downregulation of expression of *JRG21* within the JA signaling pathway was most pronounced at 3 h, with 5-fold reduction in expression relative to expression at 0 h.

#### 2.3.7. Analysis of DEGs Involved in Redox-Based Reaction Processes upon Salt Stress Exposure

A total of 24 DEGs were involved in redox-associated processes. Compared with the WT line, 11 genes were upregulated at 0 h in transgenic lines, such as *FE SUPEROXIDE DISMUTASE 1* (AT4G25100.2), *PRX9* (AT1G44970.1), *ARABIDOPSIS THALIANA FERRETIN 1* (AT5G01600.1), and *FERRITIN 4* (AT2G40300.1), among others, while 9 genes were downregulated, such as *CYTOCHROME P450* (AT3G28740.1), *BETA-CAROTENE HYDROXYLASE 2* (AT5G52570.1), *DARK INDUCIBLE 11* (AT3G49620.1), and others. After 3 h of stress exposure, 8 genes were upregulated in transgenic versus nontransgenic lines, including *FSD1* (AT4G25100.2), *PRX9* (AT1G44970.1), and others, while 16 genes were downregulated, including *POLYPEPTIDE 13 PUTATIVE CYTOCHROME P450* (AT2G30770.1), *Peroxidase 52* (AT5G05340.1), and others (Figure 8). Notably, at 3 h, transcription of the gene encoding ZINC-BINDING DEHYDROGENASE FAMILY PROTEIN (AT5G37940.1) was upregulated by as much as 7-fold relative to its expression at 0 h, while *ELI3-2* (AT4G37990.1) was downregulated by 2-fold.

#### 2.3.8. qRT-PCR Analysis

We randomly selected six genes and verified their expression levels under NaCl stress at 0 h and 3 h (Figure 9). The qRT-PCR and transcriptome data trends of *FSD1* (AT4G25100.2) and *ELICITOR-ACTIVATED GENE 3* (AT4G37990.1) exhibited the same trends (Figure 9A,D), with their expression levels decreasing after 3 h of NaCl stress. Meanwhile, *AP2* (AT4G34400.1), *PRX9* (AT1G44970.1), *SERINE CARBOXYPEPTIDASE S28* (AT4G36190.1), and CYTOCHROME P450 (AT4G37410) exhibited a shared upward expression trend (Figure 9B,C,E,F) and attained highest expression levels under NaCl stress at 3 h. These results demonstrate the reliability of the RNA-seq results.

## 3. Discussion

### 3.1. Overexpressing ThTRX5 Improves Salt Resistance of Arabidopsis Plants

Thioredoxin, which plays an important role in resistance to drought, oxidation, and heat stresses, has important roles in signal transduction and plant development. In this study, we constructed the plant expression vector pGWB5-ThTrx5 and successfully transformed it into *Arabidopsis*.

Under NaCl treatment, seed germination rates of transgenic *Arabidopsis* lines were significantly greater than the corresponding WT rate (Figure 1). After exposure to 150 mM NaCl for 48 h, leaves of the WT line yellowed, while most transgenic lines grew normally and exhibited root lengths and fresh weight values that were consistently higher than corresponding values for the WT line (Figure 2). These results suggest that salt tolerance of transgenic lines exceeded that of the nontransgenic line, with results for chlorophyll content and physiological indicators further supporting this conclusion. These results are noteworthy, as the *ThTrx5*-overexpressing transgenic lines generated here appear to have heightened resistance to salt stress that may be due to the importance of the Trx system and the glutathione-glutaredoxin (Grx) system, which together form the main antioxidant system in plants [15]. The Trx system not only has the same reducibility as the Grx system, but also reduces intracellular oxidized glutathione to maintain the intracellular reduced glutathione concentration [16]. Maintaining high reducing capacity under stress is necessary for plants to cope with adverse conditions and redox level increases [17]. Here, abundances of major antioxidant enzymes and a nonenzymatic antioxidant in plant cells (SOD, POD, CAT, and glutathione, respectively) were further examined in WT and transgenic lines. Under salt stress, levels of SOD, CAT, POD, and GSSG contents in transgenic line plants were higher than levels in WT plants (Figure 2), indicating that *ThTrx5* overexpression increased salt tolerance of *A. thaliana*. This result aligns with previously reported results demonstrating that transgenic *Arabidopsis* overexpressing *GhSnRK2* showed obvious salt tolerance that was associated with stress response processes that reduced water loss, regulated cell turgor, maintained relative water content, and controlled proline accumulation [18]. In fact, the importance of glutathione to plant salt stress responses has also been demonstrated in tomato, as glutathione content increased significantly under high salt exposure [19]. 

### 3.2. Arabidopsis Proteins Interacting with ThTRX5 Participate with Stress Resistance

To further investigate salt tolerance mechanisms associated with expression of *ThTrx5*, we used pull-down technology to identify five proteins that interact with ThTrx5: catalase 3 (CAT3), glyceraldehyde-3-phosphate dehydrogenase (GAPDH), peptidyl-prolyl cis-trans isomerase (CYP20-3), ATP synthase subunit beta, and a peroxidase, 2-Cys peroxiredoxin BAS1-like protein (Figure 3). We analyzed the domains of three *Tamarix* and four *Arabidopsis* m subclasses genes, the results showed that these seven genes all contained highly conserved WCGPC domains, and *ThTrx5* had the highest homology with *AtTRX-M4* gene of *Arabidopsis* (Appendix A). Therefore, we selected the *AtTRX-M4* gene and predicted the proteins interacting with it by using ©STRING CONSORTIUM 2019 software. The prediction results show that 2-Cys peroxiredoxin BAS1-like protein (At5g06290) can interact with it (Appendix A), and the results of BAS1 and pull-down experiments are consistent, the protein and ThTrx5 were both localized in the chloroplast. Interestingly, in the results of protein prediction, we found that AtTRX-M4 also interacted with AtTRX1, and this reaction was also performed in chloroplasts. Glycerol-3-phosphate dehydrogenase (GAPDH) was screened in the pull-down experiment. We found that under reducing conditions, AtTRX1 can activate in chloroplast the glyceraldehyde-3-phosphate dehydrogenase, so we speculated that in the chloroplast, ThTRX5 may interact with TRX1 to catalyze glyceraldehyde-3-phosphate dehydrogenase, and this reaction was under reducing conditions, it was performed in the next step, and because pull-down was an in vitro experiment, it only screened GAPDH.

Important enzymes in maintaining the chloroplast antioxidant protection are the 2-Cys peroxiredoxins [20,21], which are ubiquitous and evolutionary old enzymes reducing H_2_O_2_ and alkyl hydroperoxides [22,23]. These nuclear-encoded proteins are highly abundant within the chloroplast, where they attach to the thylakoid membrane under stress conditions [24]. Peroxiredoxins (Prxs) function both as redox sensory system within the network and as redox-dependent interactors. The processes directly or indirectly targeted by Prxs include gene expression, post-transcriptional reactions, including translation, post-translational regulation, and switching or tuning of metabolic pathways, and other cell activities. Margarete Baier et al. demonstrated that peroxiredoxins respond to ABA signals [25]. BAS1 is a 2-Cys peroxidase (Prx) in plants, scavenges reactive oxygen species (ROS) in vivo, regulates intracellular signaling, and serves as a chaperone [21,26]. BAS1 also participates in plant resistance to stress-induced damage. For example, in *Arabidopsis* exposed to salt, drought or low-temperature stresses, BAS1 was shown to be important for removal of toxic ROS, as well as for antioxidant defense and redox signaling [27].

Glycerol-3-phosphate dehydrogenase (GAPDH) is a ubiquitous enzyme involved in plant life-sustaining activities, including growth and development, as well as in stress responses to high salt or drought, where the enzyme response is very rapid. Notably, the importance of this enzyme to plant salt tolerance was demonstrated in a study showing that the NP-GAPDH gene in rice exhibited significantly increased expression under drought and high salt conditions [28].

### 3.3. ThTrx5 Participates in the Redox Process, TF Expression, and Hormone Signal Transduction

Tolerance of plants to environmental stress depends on their efficiency in sensing stress signals and in activating stress response mechanisms [29]. Plants perceive external salt stress through signal receptors on cell membranes that, through a series of signal transduction pathways and transcriptional regulation mechanisms, induce changes in gene expression downstream of salt stress receptors to control plant growth and development. Here, DEGs in transgenic and nontransgenic lines were analyzed at 0 h and 3 h of NaCl treatment, resulting in identification of 694 DEGs. Some of these DEGs were associated with redox processes, transcriptional regulation, and hormone signal transduction, suggesting that transcriptional regulation may underlie salt tolerance of *ThTrx5*-overexpressing lines. As clues to transcriptional regulation targets involved in salt tolerance, at 3 h of NaCl treatment, expression levels of peroxidase and superoxide dismutase in transgenic lines were upregulated by 500- and 20-fold, respectively; after stress exposure, expression of a calmodulin-like gene in transgenic lines was upregulated relative to WT expression (Figure 8). Thus, *ThThx5* appears to induce expression of superoxide dismutase, peroxidase, and calmodulin-like genes during the *Arabidopsis* salt stress response. Among them, *PRX9* and *FSD1* were identified and categorized according to GO classification (Figure 8). Meanwhile, Grx and Trx systems, considered parallel redox systems operating independently of one another, may also play a role in salt tolerance; here, transcriptome data indicated that during salt stress, *Grx* gene expression was downregulated in transgenic *ThTrx*-overexpressing lines relative to expression in the WT line, reflecting the complementary relationship between *Grx* and *Trx* systems.

In conclusion, the triggering of changes in *ThTrx5* expression is important for plant adaptation to salt stress. Specifically, ThTrx5 production is induced during high salt stress exposure via a specific signal-based activation mechanism. ThTrx5 then triggers regulatory pathways that rapidly and broadly alter downstream gene expression and perpetuate stress signal transmission, ultimately activating several protein kinases or phosphatases that enable plants to quickly respond to salt stress.

The molecular mechanism of salt stress-related gene expression and signal transduction is very complex and is the result of synergy among multiple genes and multiple pathways. Hormones play important regulatory roles in adaptation to environmental stresses [30] and are known to participate in early regulation of the plant salt stress response through complex interaction networks [31]. One such hormone, ABA, has been shown to have an important role in plant drought resistance, with a molecular mechanism of action that is relatively clear. However, in this study, expression changes of ABA-related gene pathways were not obvious (Table 2), indicating that *ThTrx5* did not improve plant salt tolerance via a mechanism involving ABA-related signaling pathways. Meanwhile, other studies have reported that ABA, GA, SA, Cytokinins (CK), and Indole-3-acetic acid (IAA) exhibit antagonistic effects, with ABA negatively regulating the SA-dependent defense response pathway [32], while JA and ETH often synergistically regulate plant biological responses to abiotic and biotic stresses [33]. Here, the results of DEGs analysis showed that genes involved in SA, ETH, and JA transduction pathways were most abundant, with transcript levels changing to differing degrees between 0 h and 3 h of NaCl exposure (Table 2). Thus, pathways involving ETH, JA, and SA may be responsible for salt resistance of *ThTrx5*-overexpressing transgenic lines.

Environmental stress can affect various normal physiological and metabolic processes in plants. Stress signals sensed by receptors on cell surfaces trigger associated signal transduction pathways, through which TFs regulate plant growth and development and adaptation to stress [34,35]. When gene expression levels of *ThTrx5*-overexpressing and WT lines under salt stress were analyzed, it was found that transcript levels encoding transcription factor families (such as MYB, WD40, YSL, ORG, WRKY, AP2, NAC, BHLH, etc.) were significantly induced (Figure 7). These results align with results of a previous study showing that under abiotic stress, NAM ATAF1/2/CUC2 (NAC) family gene expression increased and induced expression of other stress-related genes; in addition, overexpression of NAC TF genes significantly improved plant stress resistance [36]. As another example, under high salt, drought, and low-temperature stress, expression of the *MsNAC1* in alfalfa was upregulated then downregulated [37]. In this study, three NAC family genes exhibited different transcriptional expression levels, whereby *NAC1* was upregulated while *NAC19* and *NAC102* were downregulated (Figure 7). Another TF family, the basic/helix-loop-helix (BHLH) family, has been shown to play an important role in JA-mediated responses within abiotic stress regulatory networks [38]. Meanwhile, APETALA2/ethylene-responsive factor (AP2) family TFs also have been reported to be associated with abiotic stress resistance [39], with some members of this TF family shown to be pivotal members of JA and ETH signaling pathways that have important regulatory roles in abiotic stress adaptation [33]. Indeed, JA and ETH pathways were observed in this work to synergistically enhance salt tolerance of *ThTrx5* transgenic lines, supporting AP2 family gene involvement in salt tolerance in our model system. Finally, as yet another TF family, the WRKY family, exhibited upregulated expression in transgenic lines after salt stress (Figure 7), we speculate that WRKY TFs play regulatory roles in salt tolerance of transgenic *ThTrx5*-overexpressing lines. In fact, this result aligns with results of a previous study showing that overexpression of WRKY genes *GmWRKY13*, *GmWRKY21* and *GmWRKY54* in *Arabidopsis* led to improved stress resistance [40].

### 3.4. ThTrx5 Regulates Biological and Metabolic Pathways via Protein Interactions

Although accumulation of H_2_O_2_ during plant salt stress exposure results in cell damage, *ThTrx5*-overexpressing lines exhibited lower oxidative damage than did WT plants, a result likely attributable to CAT3 activation triggered by transmembrane flow of calcium ions and release of cytoplasmic calcium stores. It is also possible that BAS1, shown here to directly bind to ThTrx5, may participate in redox reactions and regulate the binding of CaCl_2_ to calmodulin, while also catalyzing O_2_ production, leading to a reaction between O_2_ and benzoic acid to form SA to trigger the plant defense response. Meanwhile, environmental stress may also stimulate ROS production and increase antioxidant enzyme activities that subsequently increase levels or activation of downstream signaling factors, such as cytosolic calcium ions and plasma membrane NADPH oxidase, to explain the high degree of transcriptome representation of calmodulin, NADPH, and SA signaling pathway genes observed in this study.

## 4. Materials and Methods

### 4.1. Materials

#### 4.1.1. Plant Materials and Growth Conditions

Seedlings of *Tamarix hispida* were grown in pots containing a mixture of turf peat and sand (2:1 *v*/*v*) in a greenhouse under controlled conditions of 70–75% relative humidity, a light/dark cycle of 14/10 h (with the light cycle beginning at 7:00 a.m. daily), and a constant temperature of 24 °C. Two-month-old seedlings were used for cloning and expression pattern of the *ThTrx5* gene.

*Arabidopsis thaliana* salt-sensitive strain Columbia-0 (Col-0) was used in subsequent studies. Transgenic and wild-type (WT) *Arabidopsis* seedlings were grown on 1/2 Murashige and Skoog (MS) agar plates in a climate-controlled chamber under fluorescent light (400 μmol∙m^−2^∙s; 16 h light/8 h dark) at 22 ± 2 °C, then after 8–10 d were transplanted into soil and grown in environmentally controlled growth cabinets as described above.

#### 4.1.2. Processing Method

*T. hispida* seedlings cultivated in the soil for 2 months were selected, and were irrigated with 150 mM NaCl, and then the leaves of 10 samples were harvested at 0, 3, 6, 12, 24, and 48 h of NaCl exposure, respectively. Harvested samples were frozen quickly in liquid nitrogen, then stored at 80 °C for RNA analysis.

Homozygous seeds of transgenic *ThTrx5* and wild-type *Arabidopsis* were sown on 1/2 MS medium containing 100 mM or 150 mM NaCl after vernalization, then cultured under light-controlled conditions (16 h light/8 h dark). Germination rate was recorded 7 days later for triplicate groups of samples, with each group containing 30 seeds per line. Transgenic and WT seedlings possessing two cotyledons after 7 days of germination were transferred and cultured in a climate-controlled room in 1/2 MS or 1/2 MS medium containing 150 mM NaCl for 48 h, then 30 seedlings were harvested at 0 h and 48 h in order to determine fresh weight, root length, and SOD, POD, CAT, and GSSG content values in triplicate using GSH and GSSH detection kits (Biyuntian Company) and SOD, POD, and CAT determination kits (Nanjing Jianjian Bioengineering Research Institute). Chlorophyll content was determined using the method of Marker [41] in triplicate in groups each containing 30 embryos. Plants were placed in medium containing 150 mM NaCl, then 60 samples were harvested at 0 h and 3 h of NaCl exposure. Harvested samples were frozen quickly in liquid nitrogen, then stored at 80 °C for use in pull-down and transcriptome analysis.

### 4.2. Gene Cloning and Vector Construction

#### 4.2.1. ThTrx5 Gene Cloning and Expression Vector Construction

Total RNA of roots of *T. hispida* plants was extracted using the CTAB method [42], then subjected to reverse transcription using ReverTra Ace^®^ qPCR RT Master Mix with gDNA Remover (Toyobo Biotechnology Co., Ltd., Shanghai, China). Based on the ORF sequence within the *ThTrx5* gene, primers were designed (upstream primer, 5′-AATCTATGATTGCTCCTTC C-3′; downstream primer, 5′-GCCAGCTCATTAACAACC-3′). Each PCR reaction contained 2.5 μL KOD Plus Buffer Solution, 2.5 μL dNTPs (2.0 mmol/L), 0.5 μL cDNA, upstream and downstream primers (10 μmol/L, 0.75 μL each), 1 μL MgSO_4_, and 0.5 μL KOD, with total reaction volume adjusted to 20 μL with ddH_2_O. Amplification was performed using thermal cycling steps of 1 cycle of 94 °C for 4 min, 35 cycles of (94 °C for 45 s, 58 °C for 45 s, 68 °C for 30 s), and 1 final extension step of 68 °C for 10 min. Amplified products were detected via electrophoresis on 0.8% agarose gels. After PCR product purification, Tri-Octyl Phosphine Oxide (TOPO) cloning of PCR products was carried out, followed by transformation of recombinant plasmid into *Escherichia coli* Trans5α cells using a heat shock-based method. After confirmation of the construct using DNA sequencing, the insert of the TOPO recombinant plasmid was site-specifically recombined with pGWB5 vector via an LR reaction, then the completed reaction solution was transformed into *E. coli* Trans5α. Positive clones were detected via PCR (using specific primers), followed by 0.8% agarose gel electrophoresis to screen for inserts of predicted size. *ThTrx5* is in the pGWB5 vector and is driven by the 35S promoter.

#### 4.2.2. Construction of Fusion Protein Expression Vector

Using recombinant plasmid pENTR-Trx5 as template, the target fragment was amplified using PCR using amplification conditions described above. Upstream and downstream primers were 5′-CACCCCGCGGAATCTATGATTGCTCCTTCC-3′ (*Sac* II); 5′-CTCGAGGCCAGCTCATTAACAACC-3′ (*Xho* I). Target product obtained via PCR amplification was digested with restriction endonucleases *Sac* II and *Xho* I, then ligated to *Sac* II/*Xho* I-digested expression vector pET-41a, which contains glutathione-S-transferase (GST) Tag. Ligated product was transformed into *E. coli* Trans5α competent cells, then cells were plated onto agar medium containing kanamycin (Kan, 50 μg/mL) and allowed to grow until isolated colonies were visible. DNA extracted from colonies was screened for *ThTrx5*-containing inserts, then positive colonies for inserts were cultured, and recombinant plasmid DNA extraction performed using the BioFlux Biospin Plasmid DNA Extraction Kit (Fluxion Biosciences, Alameda, CA, USA). Confirmation of recombinant plasmid pET41a-Trx5 assembly was performed using double-digestion with *Sac* II and *Xho* I, followed by agarose gel electrophoresis and DNA sequencing. Next, 1 μL of recombinant pET-41a plasmid was transformed into Rosetta-gami B(DE3) cells via heat shock, followed by growth in Luria-Bertani (LB) medium containing 30 ug/mL Kan and 34 ug/mL chloramphenicol (Cm).

#### 4.2.3. qPT-PCR

In *T. hispida*, total RNA from leaves was extracted using the CTAB method, and reverse transcribed after NaCl treatment for 0, 3, 6, 12, 24, and 48 h. Primers were designed based on the full-length cDNA sequence of the *ThTrx5* gene (Table 3 for primers). The *α-tubulin* gene was used as internal references to normalize the amount of total RNA present in each reaction.

In *Arabidopsis thaliana*, total RNA of WT and *ThTrx5*-overexpressing lines was extracted after NaCl treatment for 0 h and 3 h, followed by reverse transcription performed on all samples. Concurrently, six genes were randomly selected from transcriptome DEGs, with *α-Tubulin* serving as internal reference gene (Table 3 for primers). Expression levels of surrogate genes in transgenic and WT plants were analyzed to verify the reliability of transcriptome data.

For use as template, cDNA was diluted 10-fold in real-time quantitative PCR reactions containing 6 μL Top qMix, 0.24 μL Passive Reference Dye, 2 μL cDNA, and 0.24 μL each of 10 μmol/L upstream and downstream primers, then adjusted to a total reaction volume of 12 μL with ddH_2_O. DNA amplification was conducted using an ABIPRISM^®^ 7500 real-time PCR instrument with cycling parameters of predenaturation at 94 °C for 30 s, followed by 45 cycles of (94 °C for 5 s, 56 °C for 15 s, 72 °C for 34 s), and a final cycle of 95 °C for 15 s, 60 °C for 1 min, and 95 °C for 30 s. All samples were amplified in triplicate, and relative quantitative analysis of genes was performed using the −ΔΔCt method. The ordinate is the value taken by log_2_.

#### 4.2.4. Data Analysis

The GO analysis method is a hypergeometric test algorithm; the DEG analysis software is based on a Poisson distribution model used with the RPKM algorithm; SPSS 7.0 statistical software was used for data analysis, *p* ≤ 0.05 from a Student’s t test. All experiments were repeated three times.

### 4.3. Preparation of Engineered Bacteria and Arabidopsis Transformation

The pGWB5-Trx5 plasmid construct was transferred into *Agrobacterium tumefaciens* EHA105 by electroporation, and cells were grown on Slab LB medium containing 50 mg/mL Kan and 50 mg/mL rifampicin (Rif). A single colony was inoculated into 5 mL of Luria-Bertani (LB) medium containing antibiotics Kan and Rif, then incubated at 28 °C overnight, then cultured a second time in 50 mL of fresh LB medium without antibiotics at 28 °C overnight. *A. tumefaciens* cells were harvested after centrifugation for 15 min at 5000 rpm, resuspended in 5% sucrose water solution, adjusted to an OD_600_ of 0.6, then were used to transform *Arabidopsis* (Col-0) using a floral dip-based transformation method [43]. Transformants were selected by planting seeds onto 1/2 MS medium containing 50 mg/L Kan. Transgenic lines were confirmed using PCR, then propagated to the T_3_ generation, then rescreened to confirm successful development of stable transgenic lines (homozygous for the *ThTrx5* gene) for use in subsequent experiments.

### 4.4. Screening of Interacting Proteins

#### 4.4.1. Induction of Protein Expression

A single colony of Rosetta-gami B(DE3) cells was selected then cultured in vitro in 4 mL LB medium (containing 30 μg/mL Kan, 34 μg/mL Cm) at 37 °C for 12 h with shaking (220 rpm). Next, the culture was transferred to a culture flask containing 4 L of LB medium (30 μg/mL Kan, 34 μg/mL Cm), for a culture dilution of 1:1000. Induction was performed at 20 °C and 37 °C, respectively, and finally, 37 °C was the best induction condition. Therefore, the flask was incubated at 37 °C with shaking at 200 rpm until the OD_600_ reached 0.6. Isopropyl-β-d-thiogalactopyranoside (IPTG) was added to a concentration of 0.2 mmol/L to induce protein expression, and the flask was incubated at 37 °C with shaking (200 rpm) for 5 h. *E. coli* cells were collected, then suspended in buffer (50 mM Tris, 300 mM NaCl, 0.1% Triton X-100, pH 8.0). Cells were broken using low-temperature ultrasound for 20 min, then centrifuged for 20 min at 12000 rpm and 4 °C. The supernatant was diluted in equilibrium buffer (50 mM Tris, 100 mM NaCl, 0.1% Tween-20, 1 mM Phenylmethylsulfonyl fluoride (PMSF), 1 mM DL-Dithiothreitol (DTT) to a volume equivalent to the original volume before centrifugation, then SDS-PAGE electrophoresis and western blot were performed to analyze ThTrx fusion protein obtained using GST antibody.

#### 4.4.2. Pull-Down

The wild-type and ThTrx5 transgenic *Arabidopsis thaliana* growing for 7 d were selected, treated with 150 mmol/L NaCl for 3 h, and the leaves were collected. Total protein of transgenic and nontransgenic *Arabidopsis* lines was extracted using a One-Step Plant Protein Extraction Kit (Shanghai Shenggong Biotechnology Co., Ltd., Shanghai, China). Extracted total protein was concentrated to a volume of 400 μL using a centrifugal filter cartridge (Millipore, MA, USA; Agent: Bioflux, Biotech Innovation Technology Co., Ltd., Beijing, China). After Coomassie Brilliant Blue staining to confirm the amount of protein, approximately 50 μg of total protein from each transgenic and nontransgenic *Arabidopsis* line was incubated with nickel agarose resin beads for 4 h at 4 °C. Beads were then washed 5 times with 1 mL of elution buffer (50 mM Tris, 100 mM NaCl, 0.1% Tween-20, 1 mM PMSF, 1 mM DTT, 500 mM imidazole). SDS-PAGE was performed. The bands that appeared in transgenic plants and wild type were excised for differential bands for analysis by mass spectrometry.

#### 4.4.3. Mass Spectrometry

Mass spectrometry was conducted using an Nd: YAG laser delivering light of wavelength 355 nm using an acceleration voltage of 2 kV, with data collection using positive ion and automatic data acquisition modes. The Peptide mass fingerprinting (PMF) mass scan range was 800–4000 Da. For tandem mass spectrometry (MS/MS) analysis, eight precursor ions were selected for each sample point; secondary MS/MS laser excitation sampling was set to 2500 times, collision energy was set to 2 kV, and Collision-Induced Dissociation (CID) was turned off.

### 4.5. Transcriptome Analysis

#### 4.5.1. RNA Extraction and Differential Expression Library Construction

Total mRNA was isolated using the CTAB method, then was digested with DNase I (RNase-free) for 30 min at 37 °C to remove genomic DNA contaminants. RNA integrity was assessed using a NanoPhotometer^®^ Spectrophotometer (IMPLEN, Westlake Village, CA, USA) and RNA Nano 6000 Assay Kit, and data were analyzed using the Agilent Bioanalyzer 2100 System (Agilent Technologies, Santa Clara, CA, USA). Total mRNA (20 μg per sample) with RNA integrity (RIN) score greater than 8.0 was sent to Huada Gene Technology (Shenzhen, China) for construction of individual cDNA libraries for Illumina sequencing. RNA sequencing (RNA-seq) was performed using an Illumina kit and standard cBot System using sequencing protocols provided by the manufacturer. Briefly, mRNA was isolated using oligo (dT) cellulose and broken into short fragments by addition of fragmentation buffer. Using these short fragments as templates, first-strand cDNA and second-strand cDNA were synthesized. Sequencing adapters (used to distinguish among sequences derived from different libraries) were ligated to the short fragments after purifying fragments using a QIAQuick PCR Extraction Kit. Fragments 200 ± 25 bp in length were then separated by agarose gel electrophoresis and selected for use as sequencing templates in PCR amplification. All experiments were conducted using three biological replicates per experimental sample. Finally, the two libraries were sequenced using Illumina HiSeq™ 2000.

#### 4.5.2. De Novo Assembly and Functional Annotation

To obtain accurate data, raw reads were first filtered by removing adapter and low-quality sequences, which included sequences with high percentages of poor base calls (high N percentages of over 5%) and sequences containing more than 20% nucleotides with Q-values ≤10, where Q-value represents the sequencing quality of a given base. Only clean reads were used in de novo assembly, and read-mapping to the transcriptome. RNA-seq data were de novo assembled using the Trinity software package for RNS-seq assembly [44]. Short reads were first assembled into longer (but gapless) contigs, then additional reads were then mapped back to previously assembled contigs, with the length of a paired-end read defined as a frame. Next, contigs were connected to access end sequences that could not be further extended from either end to ultimately produce unigene sequences. Next, unigenes were further analyzed and assembled to obtain maximum length nonredundant unigenes using TGICL clustering software set to a minimum overlap length of 100 bp. Assembled genes were functionally aligned against the following public databases: KEGG (Kyoto Encyclopedia of Genes and Genomes) and GO (Gene Ontology).

## 5. Conclusions

Expression of *ThTrx5* and associated protein ThTrx can significantly improve plant salt tolerance. During salt stress, ThTrx5 interacts with BAS1 to regulate biological and metabolic pathways involving genes encoding redox processes, hormone signaling pathways, and transcription factors to improve salt tolerance of plants.

## Figures and Tables

**Figure 1 ijms-21-01165-f001:**
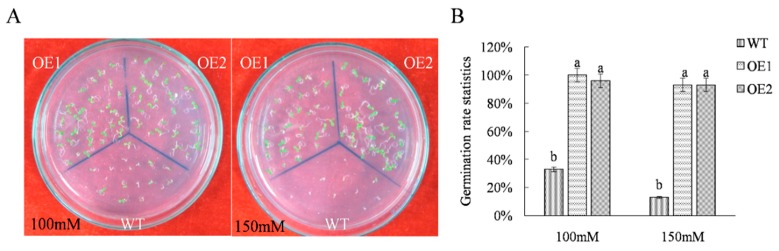
Determination of germination rates of wild-type and transgenic *Arabidopsis thaliana* under NaCl stress. (**A**) Phenotypic observation of wild-type (WT) and transgenic lines under different NaCl stress. (**B**) Statistics of germination rate of WT and transgenic *ThTrx5 Arabidopsis* seeds under NaCl stress. Three biological replicates were analyzed. Error bars show the average ± standard deviation of wild-type and transgenic lines (OE1, OE2). Different lowercase letters indicate significant differences (*p* ≤ 0.05 from a Student’s t test). WT was Columbia-0 wild-type *Arabidopsis thaliana*; OE1, OE2 was *ThTrx5*-Overexpressing Transgenic Lines.

**Figure 2 ijms-21-01165-f002:**
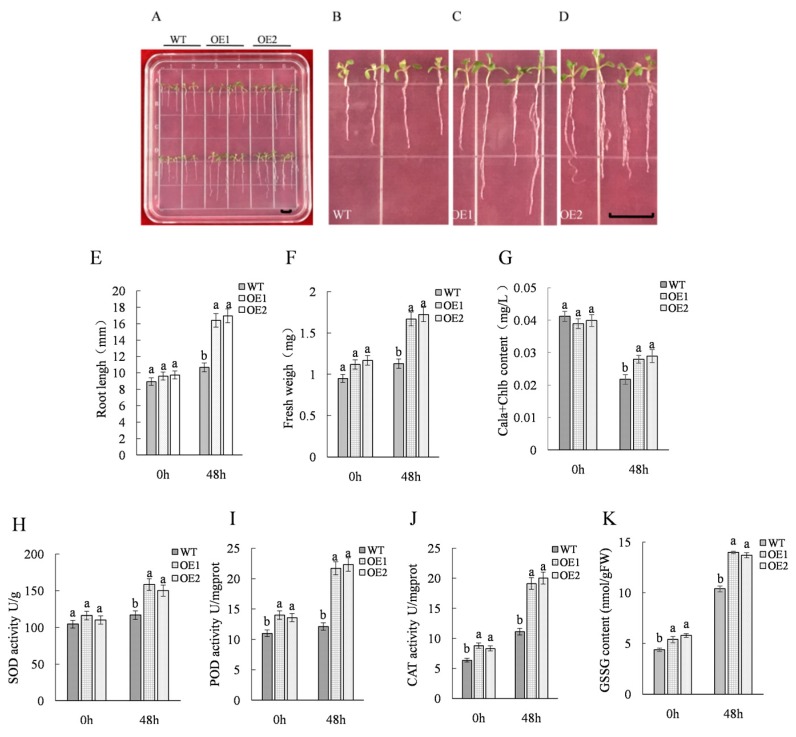
Analysis of *Arabidopsis* seedlings and determination of physiological indexes treated with 150mM NaCl. (**A**) Stress resistance analysis of *Arabidopsis thaliana* seedlings under NaCl stress. Bar 5 mm. (**B**–**D)** are enlarged views of WT, OE1, and OE2 seedlings grown under 150 mM NaCl, respectively. Bar 5 mm. (**E**) The root length of *Arabidopsis* seedlings under NaCl stress. (**F**) Fresh weight of *Arabidopsis* seedlings under NaCl stress. (**G**) Chlorophyll content in *Arabidopsis* seedlings under NaCl stress. (**H**) SOD (superoxide dismutase) content in *Arabidopsis* seedlings under NaCl stress. (**I**) POD (peroxidase) content in *Arabidopsis* seedlings under NaCl stress. (**J**) CAT (catalase) content in *Arabidopsis* seedlings under NaCl stress. (**K**) GSSG (glutathione) content in *Arabidopsis* seedlings under NaCl stress. Three biological replicates were analyzed. Error bars show the average ± standard deviation of wild-type and transgenic lines (OE1, OE2). Different lowercase letters indicate significant differences (*p* ≤ 0.05 from a Student’s t test).

**Figure 3 ijms-21-01165-f003:**
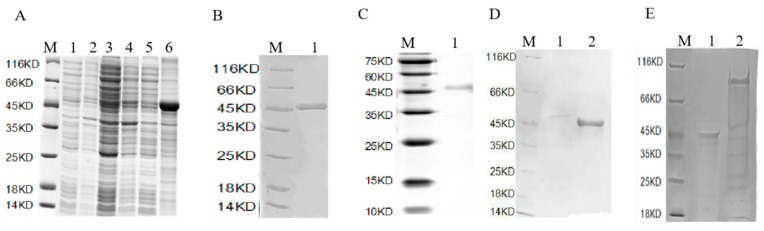
Detection of fusion expressed protein and screening of ThTrx5 interacting proteins under salt stress. (**A**) SDS-PAGE analysis of fusion protein. M: protein marker; 1: supernatant before induction; 2: precipitation before induction; 3: supernatant at 20 °C; 4: 20 °C induced precipitation; 5: 37 °C induced supernatant; 6: 37 °C induced precipitation; (**B**) SDS-PAGE analysis of fusion protein purification. M protein marker; 1: Purified protein; (**C**) Western Blot analysis of fusion protein. M: protein marker; 1: Trx5-GST protein; (**D**) Identification interaction proteins in WT by SDS-PAGE. M: protein marker; 1: column; 2: total protein of wild-type *Arabidopsis* after salt treatment; (**E**) Identification interaction proteins in transgenic *Arabidopsis* by SDS-PAGE. M: protein marker; 1: Column; 2: Total protein of transgenic ThTrx5 *Arabidopsis* after salt treatment. GST: glutathione-S-transferase.

**Figure 4 ijms-21-01165-f004:**
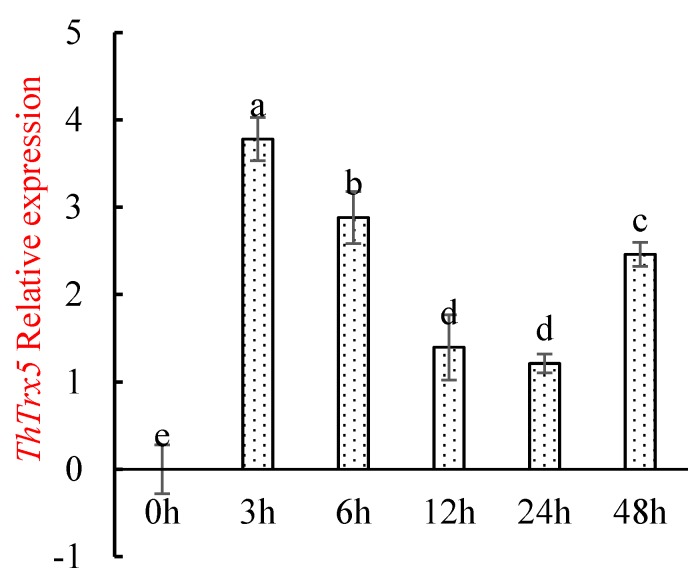
Relative expression of *ThTrx5* gene in *T. hispida* under NaCl stress. Error bars show the standard deviation of *T. hispida*. Different lowercase letters indicate significant differences (*p* ≤ 0.05 from a Student’s t test).

**Figure 5 ijms-21-01165-f005:**
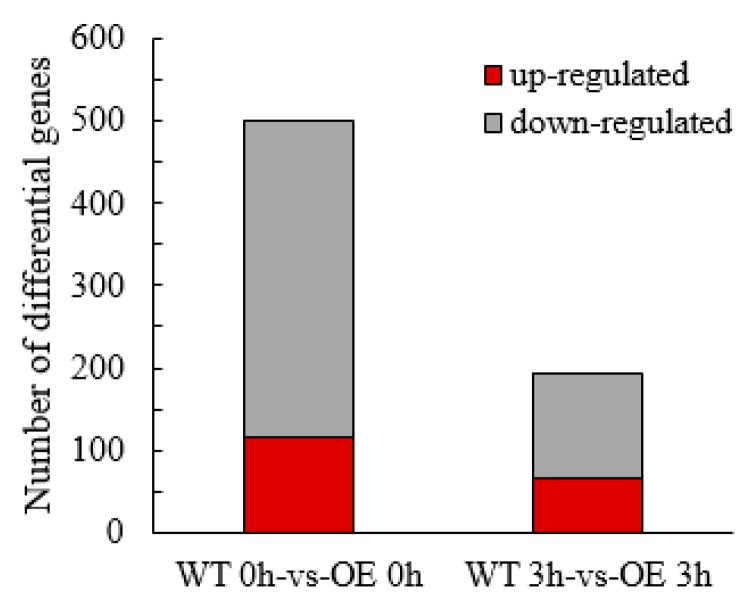
Differential gene distribution map.

**Figure 6 ijms-21-01165-f006:**
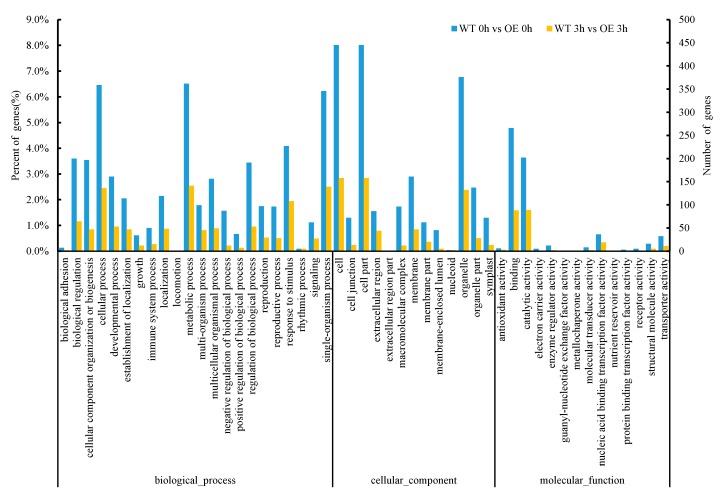
Gene Ontology (GO) classification of WT and ThTrx5 transgenic lines under salt treatment for 0 h and 3 h.

**Figure 7 ijms-21-01165-f007:**
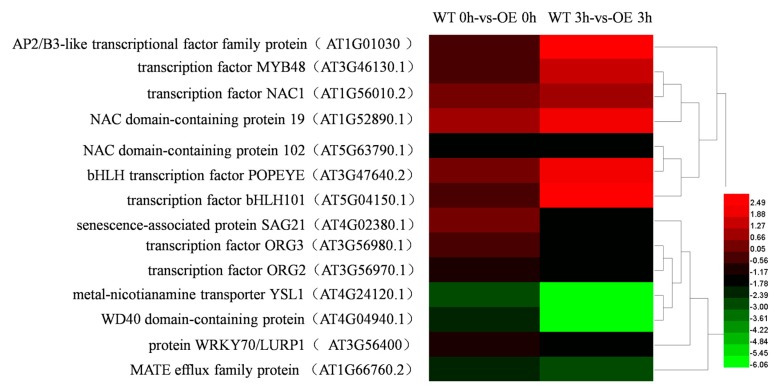
Expression analysis of differentially expressed Transcription Factors (TFs) during stress exposure.

**Figure 8 ijms-21-01165-f008:**
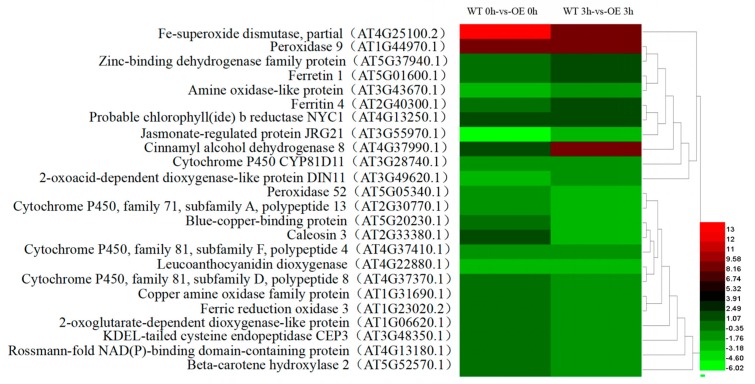
Expression analysis of DEGs involved in redox reaction processes during salt stress exposure.

**Figure 9 ijms-21-01165-f009:**
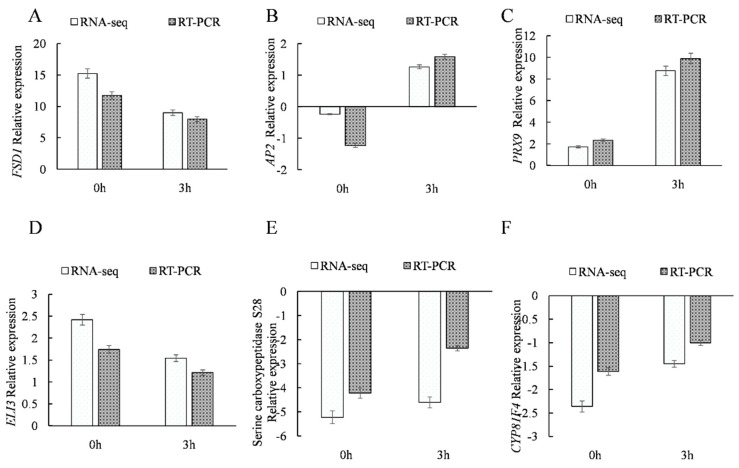
The expression levels of the genes were studied at 0 h and 3 h of exposure to 150 mM NaCl. (**A**) AT4G25100.2; (**B**) AT4G34400.1; (**C**) AT1G44970.1; (**D**) AT4G37990.1; (**E**) AT4G36190.1; (**F**) AT4G37410.

**Table 1 ijms-21-01165-t001:** Quality of RNA-Seq of ThTrx5 transgenic lines.

Sample ID	WT-0h	WT-3h	OE-0h	OE-3h
Total Reads	7,639,214(100.00%)	7,499,090(100.00%)	7,165,453(100.00%)	7,318,039(100.00%)
Total BasePairs	366,468,011(100.00%)	357,615,524(100.00%)	346,511,389(100.00%)	358,182,356(100.00%)
Total Mapped Reads	4,547,338(60.80%)	5,339,751(73.16%)	4,739,829(67.03%)	5,396,632(73.83%)
Perfect Match	4,063,215(54.33%)	4,756,366(65.17%)	4,241,363(59.98%)	4,835,471(66.15%)
<=2bp Mismatch	484,123(6.47%)	583,385(7.99%)	498,466(7.05%)	561,161(7.68%)
Unique Match	4,291,451(57.38%)	5,119,890(70.15%)	4,490,200(63.50%)	5,162,943(70.63%)
Multi-position Match	255,887(3.42%)	219,861(3.01%)	249,629(3.53%)	233,689(3.20%)
Total Unmapped Reads	2,931,601(39.20%)	1,958,525(26.84%)	2,331,832(32.97%)	1,913,212(26.17%)
Clean Reads	7,478,939	7,298,276	7,071,661	7,309,844

**Table 2 ijms-21-01165-t002:** Expression analysis of DEGs involved in hormone synthesis and signal transduction pathways during salt stress exposure.

Hormone	Gene	Log_2_ (WT 0 h/Trx 0 h)	Log_2_ (WT 3 h/Trx 3 h)	Gene Annotation
ABA	AT1G52040.1	−1.466825674	−1.650566816	myrosinase-binding protein 1
BR	AT4G34650.1	0.06962026	−1.262992532	squalene synthase 2
ETH	AT5G47220.1	−0.468665009	1.963721496	ethylene-responsive transcription factor 2
	AT2G20880.1	0.519653186	−1.327806826	ethylene-responsive transcription factor ERF053
	AT1G69260.1	−0.387237409	−1.327806826	ABI five binding protein
JA	AT3G55970.1	−0.457626737	−2.592176442	jasmonate-regulated protein JRG21
	AT3G50760.1	0.502974445	−1.129387909	Probable galacturonosyltransferase-like 2
	AT5G24780.1	−2.57480625	−2.023514997	acid phosphatase VSP1
	AT5G24770.1	−0.608451639	−1.443297712	acid phosphatase VSP2
	AT2G39030.1	−3.23523431	−2.243926852	L-ornithine N5-acetyltransferase NATA1
6-BA	AT1G21310.1	−0.4968619	1.346863463	response to cytokinin stimulus
SA	AT3G20470.1	0.527147723	1.015449778	glycine-rich protein 5

**Table 3 ijms-21-01165-t003:** PCR primer sequences.

Primer Name	Forward Primer(5′→3′)	Reverse Primer(5′→3′)
*α-Tubulin*	5′-GCACTGGCCTCCAAGGAT-3′	5′-TGGGTCGCTCAATGTCAAGG-3′
*ThTrx5*	5′-AATCTATGATTGCTCCTTCC-3′	5′-GCCAGCTCATTAACAACC-3′
AT4G25100.2	5′-GATGCTTTGGAGCCGCATATG-3′	5′-GAAGAACTCGTGGTTCCACG-3′
AT4G34400.1	5′-CAGCTCAGAGTTCATGGTGATC-3′	5′-GTGAGCTCCATTATAGGCAAAGG-3′
AT1G44970.1	5′-CGTGAGTGGATTCCCTAAACAATC-3′	5′-CACTCCAAACCCAGAGATTGG-3′
AT4G37990.1	5′-CGAAAGACAATTCCGGAGTTCTC-3′	5′-CAGTCACCACGCCCACGATC-3′
AT4G36190.1	5′-GGTTACTGCAACCTCGTAGGATTTC-3′	5′-GGTGATTCCATTGCAAGGTCC-3′
AT4G37410	5′-CCTCATCAAACCGCCGGTTC-3′	5′-AGCAATGTATTTGGAGGTTAGAAAACG-3′

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
