# Peer review of "Overexpression of Tamarix hispida ThTrx5 Confers Salt Tolerance to Arabidopsis by Activating Stress Response Signals"

_ijms, 2020, doi:10.3390/ijms21031165_

Round 1
Reviewer 1 Report
The authors have acceptably addressed most of the questions raised in my previous review. The changes made by them have substantially improved the first version of the manuscript. Notwithstanding, some comments I made in my former revision have not been addressed. In addition, there are several aspects that require further clarification as well as some mistakes that should be corrected, for the paper to be suitable for publication.
Lines 88-89. It can be read “leaves of WT plants were obviously chlorotic and yellow, while most plants of OE1/OE2 lines grew normally (Figure 2A)”. However, this can not be concluded from Fig. 2A. Plants are too small. A close-up view of the plants clearly showing the leaf yellowish phenotype is required.
The Arabidopsis nomenclature has not been unified. It can still be read Arabidopsis thaliana (in italics), Arabidopsis (in italics), A. thaliana and Arabidopsis (without italics). This must be done.
Figures 2B and C. What does "Relatively" mean in the labels of the axes?
Figure 2C. Units for the “y” axe are missing.
Figure 2 legend. Indicate the NaCl concentration used. Moreover, references to Fig. 2B and 2C have been wrongly exchanged.
Figure 3 legend. According to the information in Material and Methods, lines 422-423, the wild type and OE lines used for pull-down assays, were untreated. However, here it can be read "Screening interaction protein of ThTrx5 under salt stress". This is contradictory.
Figure 3 legend. Correct “column boun”. Should be “Column-bound proteins”.
Figure 4. Which OE line was studied? In the text it can be read "transgenic lines" but apparently only one OE is shown. This must be clarified.
Section 2.3.1 to 2.3.4. The way fold changes in gene expression levels are described in the text is bizarre and I do not think is correct. For instance in line 135, it can be read “For transgenic lines, ThTrx5 gene expression reached its highest value 211 fold-higher expression at 3 h than at 0 h”. But taking a look to Fig. 4, ThTrx5 gene expression levels was around 2 times higher in the OE line at 3 h than at 0 h and not 211 times higher. This requires a revision and correction.
Figure 4. Indicate that gene expression levels were determined by qRT-PCR as well as the housekeeping (control) gene used.
Section 2.3.6, 2.3.7 and 2.3.8. Full names of genes should be shown the first time they are mentioned in the manuscript.
Lines 270-271. Correct the font for “…generated here appear to have heightened resistance to salt stress that may be due to the importance of the”
Author Response
Response to Reviewer 1 Comments
Point 1: Lines 88-89. It can be read “leaves of WT plants were obviously chlorotic and yellow, while most plants of OE1/OE2 lines grew normally (Figure 2A)”. However, this can not be concluded from Fig. 2A. Plants are too small. A close-up view of the plants clearly showing the leaf yellowish phenotype is required.
Response 1: We have made changes as shown in Figure 2 B C D.
Point 2: The Arabidopsis nomenclature has not been unified. It can still be read Arabidopsis thaliana (in italics), Arabidopsis (in italics), A. thaliana and Arabidopsis (without italics). This must be done.
Response 2: We are very sorry for the negligence of the question. We have made correction according to the reviewer’s comments.
Point 3: Figures 2B and C. What does "Relatively" mean in the labels of the axes?
Response 3: We have carefully studied your comments and have made changes.
Point 4:Figure 2C. Units for the “y” axe are missing.
Response 4: We are very sorry for the negligence of Figure2C.We have modified it.
Point 5:Figure 2 legend. Indicate the NaCl concentration used. Moreover, references to Fig. 2B and 2C have been wrongly exchanged.
Response 5: We used 150 mM NaCl to treat Arabidopsis seedlings and analyze their physiological parameters.Already modified in the article see line 108 for details.We are very sorry for the negligence of Figure2B and Figure2C.We have made correction.
Point 6:Figure 3 legend. According to the information in Material and Methods, lines 422-423, the wild type and OE lines used for pull-down assays, were untreated. However, here it can be read "Screening interaction protein of ThTrx5 under salt stress". This is contradictory.
Response 6: We are very sorry for the negligence of Figure 3 legend.We have deleted the wrong word.
Point 7 :Figure 3 legend. Correct “column boun”. Should be “Column-bound proteins”.
Response 7:We have made correction according to the reviewer’s comments.
Point 8 :Figure 4. Which OE line was studied? In the text it can be read "transgenic lines" but apparently only one OE is shown. This must be clarified.
Response 8:We selected the OE1 strain for this experiment.Article has been changed again.
Point 9 :Section 2.3.1 to 2.3.4. The way fold changes in gene expression levels are described in the text is bizarre and I do not think is correct. For instance in line 135, it can be read “For transgenic lines, ThTrx5 gene expression reached its highest value 211 fold-higher expression at 3 h than at 0 h”. But taking a look to Fig. 4, ThTrx5 gene expression levels was around 2 times higher in the OE line at 3 h than at 0 h and not 211 times higher. This requires a revision and correction.
Response 9:In the article we have deleted the wrong data. See the article for details 2.3.1. Changes in ThTrx5 gene expression in WT and transgenic lines under NaCl stress.
Point 10 :Figure 4. Indicate that gene expression levels were determined by qRT-PCR as well as the housekeeping (control) gene used.
Response 10:The housekeeping (control) gene used was α-Tubulin.See lines 156 for details.
Point 11 :Section 2.3.6, 2.3.7 and 2.3.8. Full names of genes should be shown the first time they are mentioned in the manuscript.
Response 11:We have made correction according to the reviewer’s comments.
Point 12 :Lines 270-271. Correct the font for “…generated here appear to have heightened resistance to salt stress that may be due to the importance of the”
Response 12:We have made correction according to the reviewer’s comments.
Thanks again for your valuable feedback.
Reviewer 2 Report
The manuscript by Luan and collaborators deal with the role of a thioredoxin from Tamarix hispida.
Most of the results are based on Arabidopsis plants overexpressing the T. hispida genes.
The strategy of heterologous overexpression can be criticized because it does not really help deciphering the role of a gene. A loss of function strategy is best fitted to that.
There is no data pertaining to the Tamarix hispida plants. So the paper is about the effect of the overexpression of a TRX gene in Arabidopsis. We do not even know where the protein goes (subcellular compartment).
,
In the abstract, the fact that the gene ThTRX is a thioredoxin is not indicated. It is not indicated if this TRX is a plastidic one, cytosolic or mitochondrial one. This info should also be in the title. Or at least indicate which classes of TRX it corresponds to. Is it h, f, m, o, x or y. (it is a class I TRX, but which subclasses)?
Idem, in the introduction, a description of ThTRX5 should be documented; again, which subclass?
I think the subcellular localization of the protein should be investigated.
Lines 98/100 and 102/109:please indicate number of replicates, statistical test performed, significance of the letters.
Line 109: this is not a title!
Line 116: you write about “differential band”? Do you mean different bands? Besides I only see one? Lane 1 of B panel is bad.
You cannot present panel B before panel A. Please switch the panels. Or explain in the text panel A first.
Line 127= bound/
Line 128: this is not a title!
Line 129 and below; what is the promoter used for the expression of the gene? The native promoter from Tamaris? In section 4.2.1 it is not written, neither in the result section. In the lines 72-75; it is only mentioned ORF. So what about the promoter?
Line 80 : how are you sure the expression you measure is ThTRX5 and not an Arabidopsis orthologous
Line 132 and after: why do you use exponents of 2??? Write 8 and not 2^3
Figure 4: is the axis a log 2 ratio?
It is not indicated. What is used for control? WT at time 0. Please indicate
Line 142: here again indicate the number of replicates, statistical test performed, significance of the letters.
Line 147 and after. Why use “CK” for abbreviation of “control” . in “control” there is no K letter so it is not at all clear. Besides, CK is the abbreviations used for cytokinins for people working on hormones. So do not use CK. In your case, why do not you use WT???
Figure 6: what is interesting is not the number of genes but the enrichment ratipon of a term compared to the whole genome.
Figure 7: the colorcode is not legible
Table 2 : what is the use of so many decimals? XWhy do not you prepare a heatmap as in figures 7 and 8?
Line 259= Overexpressing ThTRX5 improves salt resistance of Arabidopsis plants
Line 286 Arabidopsis proteins interacting with ThTRX5 participate with stress resistance
You should discuss the subcellular compartment of the interacting proteins? Are they from only one compartment? Or many? If many, does it make sense?...
In conclusion, more data are necessary
1-localisation of the protein overexpressed in arabiodpsis
2- better characterization of the gene: at least mention to which subclass it is. The % of identity with Arabidopsis counterparts etc.
3-what is the promotor driving the expression?
Author Response
Response to Reviewer 2 Comments
Point 1 :In the abstract, the fact that the gene ThTRX is a thioredoxin is not indicated. It is not indicated if this TRX is a plastidic one, cytosolic or mitochondrial one. This info should also be in the title. Or at least indicate which classes of TRX it corresponds to. Is it h, f, m, o, x or y. (it is a class I TRX, but which subclasses)?
Response 1:In the foreword, we supplemented the category of TRX, which belongs to the m subclass and is located in the chloroplast.See lines 61 for details.
Point 2 :Idem, in the introduction, a description of ThTRX5 should be documented; again, which subclass?I think the subcellular localization of the protein should be investigated.
Response 2:We are very sorry that we have not performed subcellular localization experiments due to the current experimental conditions, but we did the prediction of the protein of the ThTrx5 homologous AtTHM4 and showed the localized organelles.
Point 3 : Lines 98/100 and 102/109:please indicate number of replicates, statistical test performed, significance of the letters.
Response 3:We have added in the article see lines 511 to 512 for details.
Point 4 : Line 109: this is not a title!
Response 4:We have replaced it with Screening of interacting proteins.
Point 5 :Line 116: you write about “differential band”? Do you mean different bands? Besides I only see one? Lane 1 of B panel is bad.
Response 5:We have improved the sharpness of the original picture(now Figure 3A).
Point 6 :You cannot present panel B before panel A. Please switch the panels. Or explain in the text panel A first.
Response 6:We have made correction according to the reviewer’s comments that have switched the panels.
Point 7 :Line 127= bound/
Response 7:We have made changes again.
Point 8 :Line 128: this is not a title!
Response 8:We have replaced it with Transcriptome analysis.
Point 9 :Line 129 and below; what is the promoter used for the expression of the gene? The native promoter from Tamaris? In section 4.2.1 it is not written, neither in the result section. In the lines 72-75; it is only mentioned ORF. So what about the promoter?
Response 9: 35S is the promoter used for the expression of the gene. Added in the article see lines 74 for details;4.2.1 has been written in the results section see line 118 for details;Tamarix genome information is not available or Tamarix genome is not sequenced, so cloning and analysis of Tamarix TRX5 promoter is difficult to achieve.
Point 10 :Line 80 : how are you sure the expression you measure is ThTRX5 and not an Arabidopsis orthologous
Response 10:We used ThTrx5 primers to perform qRT-PCR to confirm that the expression of ThTRX5 was determined.
Point 11 :Line 132 and after: why do you use exponents of 2??? Write 8 and not 2^3
Response 11:We have made correction according to the reviewer’s comments.
Point 12 :Figure 4: is the axis a log 2 ratio?
Response 12:.The ordinate is the value taken by log2.
Point 13 :It is not indicated. What is used for control? WT at time 0. Please indicate
Response 13: the housekeeping (control) gene was WT 0h.see lines 156 for details.
Point 14 :Line 142: here again indicate the number of replicates, statistical test performed, significance of the letters.
Response 14:We have added in the article see lines 511 to 512 for details.
Point 15 :Line 147 and after. Why use “CK” for abbreviation of “control” . in “control” there is no K letter so it is not at all clear. Besides, CK is the abbreviations used for cytokinins for people working on hormones. So do not use CK. In your case, why do not you use WT???
Response 15:We are very sorry for the negligence of the word.We have made correction according to the reviewer’s comments.
Point 16 :Figure 6: what is interesting is not the number of genes but the enrichment ratipon of a term compared to the whole genome.
Response 16:We are very sorry for the negligence of the Figure.We have made correction according to the reviewer’s comments.
Point 17 :Figure 7: the colorcode is not legible
Response 17:We have fixed the unclear colorcode.
Point 18 :Table 2 : what is the use of so many decimals? XWhy do not you prepare a heatmap as in figures 7 and 8?
Response 18: The heat map is not used in Table 2 because the heat map cannot clearly indicate different genes under the same hormone than the table.But we will attach the drawn heat map.
Point 19 :Line 259= Overexpressing ThTRX5 improves salt resistance of Arabidopsis plants
Response 19:We have made correction according to the reviewer’s comments.
Point 20 :Line 286 Arabidopsis proteins interacting with ThTRX5 participate with stress resistance
Response 20:We have made correction according to the reviewer’s comments.
Point 21 :You should discuss the subcellular compartment of the interacting proteins? Are they from only one compartment? Or many? If many, does it make sense?...
Response 21:The proteins selected by prediction and pull-down were derived from chloroplasts.
Point 22 :In conclusion, more data are necessary
localisation of the protein overexpressed in arabiodpsis
2- better characterization of the gene: at least mention to which subclass it is. The % of identity with Arabidopsis counterparts etc.
what is the promotor driving the expression?
Response 22:Thank you very much for your valuable feedback. We supplemented the data, added the analysis of the domain and the predicted map of the interaction protein of the ThTrx5 homologous gene and indicated the organelles in which it is located. We are very sorry that we cannot perform subcellular localization experiments because of insufficient experimental conditions . In this paper, we have supplemented the subclasses of the ThTrx5 gene, and the domain analysis indicates that ThTrx5 belongs to the m subclass.
Thanks again for your valuable feedback.
Round 2
Reviewer 1 Report
The authors have followed most of my suggestions but I still have several concerns that, in my opinion, make the manuscript unsuitable for publication in its current form.
Minor concerns
Line 34. Indicate “Hereinafter Arabidopsis”, after Arabidopsis thaliana.
Line 64. Remove “thaliana”.
Figure 9 legend. I suggest adding “The expression levels of the genes were studied at 0 h and 3 h of exposure to 150 mM NaCl”.
Full names for Arabidopsis proteins or genes must be in uppercase letters and not in lowercase letters (e.g. AT3G50770 gene encodes CALMODULIN-LIKE 41 and not Calmodulin-like 41).
Supplementary Figure 3. Replace “Boxes were WCXXC domains” with “The box highlights the WCXXC domains”.
Major concerns
Figures 1 and 2:
In the last version of the manuscript, it can be read “Different lowercase letters indicate significant differences in gene expression at different tissue sites (P< 0.05)”. This sentence was not reported in the previous version. However, this text is wrong, because the authors have not determined “differences in gene expression at different tissue sites” in the results corresponding to these Figures. They have measured several parameters (e.g. germination rates, root length, fresh weight or chlorophyll content) as well as different enzymatic activities. Moreover, they do not report which statistical test was used (Student´s t test maybe?). This should be corrected, and the wright sentence included at the end of the legends of Figures 1 and 2.It is not indicated which values are represented in the graphs 1B and 2E-K. Are these values the average ± standard deviation of different samples for a given genotype? This should be clearly specified in the legend of both figures.
Bar scales are missing in the pictures of both Figures.
Replace “B、C、D was an enlarged view of WT、 OE1、OE2 under 150mM NaCl stress,respectively” with “B, C and D are enlarged views of WT, OE1 and OE2 seedlings grown under 150 mM NaCl, respectively”.
Graph 2F: units of fresh weight are still missing.
Section 2.3.1 is still confusing.
The first sentence “The gene expression levels were determined by qRT-PCR and the housekeeping (control) gene was WT 0h” is wrong. The relative values of gene expression from 3h-48 h are compared with those at 0 h for WT or OE1 lines. Hence, WT 0 h cannot be a “housekeeping (control) gene”.Lines 157-158 can be read”…the ThTrx5 expression level decreased to its lowest expression at 12 h then increased by 24 h (<2-fold) and remained slightly elevated at 48 h”. However, ThTrx5 expression in WT reached the lowest level at 12 and 24h (nearly the same according to Fig. 4) and then increases reaching at 48 h nearly 2 fold times the level found at 24 h.
Lines 158-159. It can be read “For OE1 transgenic lines, ThTrx5 gene expression reached its highest value of 4-fold higher expression at 3 h than at 0 h”. However, for the OE1 line at 3h, the level was approximately twice of that at 0 h.
4 legend. Only the OE1 line was studied. Hence, change “transgenic lines” with “transgenic line OE1”.
Author Response
Response to Reviewer 1 Comments
Minor concerns
Point 1: Line 34. Indicate “Hereinafter Arabidopsis”, after Arabidopsis thaliana.
Response 1: We have made correction according to the reviewer’s comments.
Point 2: Line 64. Remove “thaliana”.
Response 2: We have made correction according to the reviewer’s comments.
Point 3: Figure 9 legend. I suggest adding “The expression levels of the genes were studied at 0 h and 3 h of exposure to 150 mM NaCl”.
Response 3: We have made correction according to the reviewer’s comments.
Point 4: Full names for Arabidopsis proteins or genes must be in uppercase letters and not in lowercase letters (e.g. AT3G50770 gene encodes CALMODULIN-LIKE 41 and not Calmodulin-like 41).
Response 4: We have made correction according to the reviewer’s comments.
Point 5: Supplementary Figure 3. Replace “Boxes were WCXXC domains” with “The box highlights the WCXXC domains”.
Response 5: We have made correction according to the reviewer’s comments.
Major concerns
Point 1: Figures 1 and 2:
In the last version of the manuscript, it can be read “Different lowercase letters indicate significant differences in gene expression at different tissue sites (P< 0.05)”. This sentence was not reported in the previous version. However, this text is wrong, because the authors have not determined “differences in gene expression at different tissue sites” in the results corresponding to these Figures. They have measured several parameters (e.g. germination rates, root length, fresh weight or chlorophyll content) as well as different enzymatic activities. Moreover, they do not report which statistical test was used (Student´s t test maybe?). This should be corrected, and the wright sentence included at the end of the legends of Figures 1 and 2.
Response 1: We are very sorry for this oversight, we have corrected the wrong sentences below Figure 1 and Figure 2 and show the method of inspection. See lines 105 to 107 and lines 118 to 120 for details.
Point 2: It is not indicated which values are represented in the graphs 1B and 2E-K. Are these values the average ± standard deviation of different samples for a given genotype? This should be clearly specified in the legend of both figures.
Response 2: Thank you very much for your valuable feedback, we have fixed this issue.See lines 105 to 107 and lines 118 to 120 for details.
Point 3: Bar scales are missing in the pictures of both Figures.
Response 3: We have made correction according to the reviewer’s comments. See lines 112 to 113 for details.
Point 4: Replace “B、C、D was an enlarged view of WT、 OE1、OE2 under 150mM NaCl stress,respectively” with “B, C and D are enlarged views of WT, OE1 and OE2 seedlings grown under 150 mM NaCl, respectively”.
Response 4: We have made correction according to the reviewer’s comments.
Point 5: Graph 2F: units of fresh weight are still missing.
Response 5: We have made correction according to the reviewer’s comments.
Point 6: Section 2.3.1 is still confusing.
Response 6: We have made the changes again.
Point 7: The first sentence “The gene expression levels were determined by qRT-PCR and the housekeeping (control) gene was WT 0h” is wrong. The relative values of gene expression from 3h-48 h are compared with those at 0 h for WT or OE1 lines. Hence, WT 0 h cannot be a “housekeeping (control) gene”.
Response 7: We have made the changes again. See lines 154 for details.
Point 8: Lines 157-158 can be read”…the ThTrx5 expression level decreased to its lowest expression at 12 h then increased by 24 h (<2-fold) and remained slightly elevated at 48 h”. However, ThTrx5 expression in WT reached the lowest level at 12 and 24h (nearly the same according to Fig. 4) and then increases reaching at 48 h nearly 2 fold times the level found at 24 h.
Response 8: We have reworked this part because another external expert questioned this, so we only selected ThTrx5 expression in wild-type lines under salt stress. See lines 152 to 158 for details.
Point 9: Lines 158-159. It can be read “For OE1 transgenic lines, ThTrx5 gene expression reached its highest value of 4-fold higher expression at 3 h than at 0 h”. However, for the OE1 line at 3h, the level was approximately twice of that at 0 h.
Response 9: Thank you very much for your comments. We have deleted this part of the data because another external audit questioned it.
Point 10: 4 legend. Only the OE1 line was studied. Hence, change “transgenic lines” with “transgenic line OE1”.
Response 10: We have changed them to relative expression of ThTrx5 gene in WT lines under NaCl stress.See lines164 for details.
Thanks again for your suggestion.
Reviewer 2 Report
Line 3 : I am not sure « directly” is relevant
Line 15 be precise : replace transgenic by overexpressing
Line 23: delete ‘directly’
Line 23: the BAS1-like protein regulates genes? What do you mean exactly??
Line 27: I do not understand your keywords: why Betula? Why no thioredoxin??
Line 61 : you say WCXXC is characteristic of m-class? But line 31 you say WCGPC is characteristic of all thioredoxins??? How is it possible that you are more precise to define all TRXs versus one subclass. So once again, how did you show it was a m TRX? Structural identities?
Line 74 “35S is the promotor” may be is better after the sentence beginning by “Next…”.
The 35S is in PGWB5 vector, right? (you should precise it in the M&M)
Line 76: replace “pollen tube pathway” by “floral dipping”
Line 123: sentence not clear : “detection of partners detected…”!
Why have you used GST antibody? The fusion protein was fused to Histag and GST? It should be precised line 118
Line 125 : why “differentially expressed”.There is no expression data here. It is very confusing.
Line 120 : why do you use total proteins from WT and transgenic Arabidopsis?
Does figure 3B correspond to the western of 3A? Why for western only one column? What do you call “target protein”. As I wrote previously, you mentioned many bands or proteins but I see only one band. Explain. What is the band you see in 3B:
Line 136 : english not good
Line 137-139: grammatically not correct.
Line 144 : not clear What do you mean by “The prediction results show that the interaction with AtTRX-M4 in Arabidopsis 11 made protein”????. All this paragraph should be rewritten. The first part of it should be in introduction and the last part in Discussion.
Figure 4: what does that mean??? The ThTRX5 is under the expression of 35S. So what do you search by this experiment?? To decipher the activity of 35S??
Besides I still consider a localization of the overexpressd proteins is necesary.
Author Response
Response to Reviewer 2 Comments
Point 1:Line 3 : I am not sure « directly” is relevant
Response 1: We have made correction according to the reviewer’s comments.
Point 2:Line 15 be precise : replace transgenic by overexpressing
Response 2: We have made correction according to the reviewer’s comments.See lines 15 for details.
Point 3:Line 23: delete ‘directly’
Response 3: We have made correction according to the reviewer’s comments.
Point 4:Line 23: the BAS1-like protein regulates genes? What do you mean exactly??
Response 4: We have deleted the wrong word. See lines 23 for details.
Point 5:Line 27: I do not understand your keywords: why Betula? Why no thioredoxin??
Response 5: I am deeply sorry for my negligence. We have added the right keywords.
Point 6:Line 61 : you say WCXXC is characteristic of m-class? But line 31 you say WCGPC is characteristic of all thioredoxins??? How is it possible that you are more precise to define all TRXs versus one subclass. So once again, how did you show it was a m TRX? Structural identities?
Response6: We have revised this part. See lines 30 and line 58 to 64 for details.
Point 7:Line 74 “35S is the promotor” may be is better after the sentence beginning by “Next…”.
Response 7: We have made correction according to the reviewer’s comments.
Point 8:The 35S is in PGWB5 vector, right? (you should precise it in the M&M)
Response 8: The 35S is in pGWB5 vector. We have added in the article. See line 485 for details.
Point 9:Line 76: replace “pollen tube pathway” by “floral dipping”
Response 9: We have made correction according to the reviewer’s comments.
Point 10:Line 123: sentence not clear : “detection of partners detected…”!
Response 10: We have reworked the section 2.2.1.See lines 30 and line 124 to 131 for details.
Point 11:Why have you used GST antibody? The fusion protein was fused to Histag and GST? It should be precised line 118
Response 11: Yes, this vector contains two tags. In order to avoid ambiguity, we have modified it in the text and unified it as GST.
Point 12:Line 125 : why “differentially expressed”.There is no expression data here. It is very confusing.
Response 12: Thank you very much for your valuable feedback. And we have reworked this part.
Point 13: Line 120 : why do you use total proteins from WT and transgenic Arabidopsis?
Response 13: We extracted the total proteins of wild-type and transgenic lines, respectively, and passed the columns, respectively.
Point 14: Does figure 3B correspond to the western of 3A? Why for western only one column?
Response 14: We have reworked this part.
Point 15: What do you call “target protein”. As I wrote previously, you mentioned many bands or proteins but I see only one band. Explain. What is the band you see in 3B:
Response 15: We have removed ambiguous words, and we have re-added the corresponding pictures (Figure 3).
Point 16: Line 136 : english not good
Response 16: We have deleted this sentence.
Point 17:Line 137-139: grammatically not correct.
Response 17: We have rewritten these words. See lines139 to 147 for details.
Point 18:Line 144 : not clear What do you mean by “The prediction results show that the interaction with AtTRX-M4 in Arabidopsis 11 made protein”????. All this paragraph should be rewritten. The first part of it should be in introduction and the last part in Discussion.
Response 18:Thank you very much for your suggestion, we have made correction according to the reviewer’s comments. See lines 331 to 340 for details.
Point 19:Figure 4: what does that mean??? The ThTRX5 is under the expression of 35S. So what do you search by this experiment?? To decipher the activity of 35S??
Response 19:We have deleted the expression levels of the transgenic lines in Figure 4.
Point 20: Besides I still consider a localization of the overexpressd proteins is necesary.
Response 20:Thank you very much for your suggestion. And this part was not included in our experiments, but it has been reported that the m subclass of TRX is localized in the chloroplast. References were 12 and 13.
Thanks again for your suggestion.
Round 3
Reviewer 1 Report
The authors have addressed the questions I raised. I have a few comments to improve the manuscript.
line 34. “Hereinafter” should not be in italics.
Section 2.3.1:
There are still errors and inaccuracies in the modifications made by the authors. For instance, in lines 149-151 it can be read: “To speculate when the ThTrx5 gene responded most significantly under salt stress, we determined the gene expression levels were determined by qRT-PCR and the reference gene was α150 –Tubulin”. I suggest replace this sentence with this one: “To determine when ThTrx5 gene responded most significantly under salt stress, we studied by qRT-PCR ThTrx5 and α-Tubulin transcript levels at 0, 3, 6, 12, 24 and 48 h of salt treatment”.
Besides, in this section, only one WT line was studied (Col-0). Hence, “WT lines” should be replaced with “WT” or “Col-0”.
Remove Chinese letters from the y axis of Fig. 4.
Author Response
Response to Reviewer 1 Comments
Point 1: line 34. to Reviewer 1 Commentsot be in italics.
Response 1: We have made correction according to the reviewer’s comments.
Point 2: Section 2.3.1: There are still errors and inaccuracies in the modifications made by the authors. For instance, in lines 149-151 it can be read: To speculate when the ThTrx5 gene responded most significantly under salt stress, we determined the gene expression levels were determined by qRT-PCR and the reference gene was α150 –Tubulin”. I suggest replace this sentence with this one: “To determine when ThTrx5 gene responded most significantly under salt stress, we studied by qRT-PCR ThTrx5 and α-Tubulin transcript levels at 0, 3, 6, 12, 24 and 48 h of salt treatment”.
Response 2: We have made correction according to the reviewer’s comments. See lines 152 to 153 for details.
Point 3: Besides, in this section, only one WT line was studied (Col-0). Hence, 1T lines” should be replaced with “WT” or “Col-0”.
Response 3: We have made correction according to the reviewer’s comments. We have replaced “WT lines” with “T. hispida”.
Point 4:Remove Chinese letters from the y axis of Fig. 4.
Response 4:We are very sorry for this oversight, we have corrected the wrong sentences below Figure 4.
Reviewer 2 Report
Authors took into consideration some of my previous remarks
Unfortunately, I still have comments
Line 60 : found ant not founded
Line 63 : a space before “The results…”
Line 64 : homology or identity
Line 60 : found ant not founded
Line 63 : a space before “The results…”
Line 64 : homology or identity
I still do not understand figure 6D and E
Figure 4: is it WT Arabidopsis? How can you check the expression of Tamaris gene in Arabidopsis?
I still do not understand figure 6D and E
Figure 4: is it WT Arabidopsis? How can you check the expression of Tamaris gene in Arabidopsis?
Author Response
Response to Reviewer 2 Comments
Point 1: Line 60: found ant not founded
Response 1: We have made correction according to the reviewer’s comments.
Point 2: Line 63: a space before “The results…”
Response 2: We have made correction according to the reviewer’s comments.
Point 3: Line 64: homology or identity
Response 3: We have changed homology to identity.
Point 4: Line 60: found ant not founded
Response 4: We have made correction according to the reviewer’s comments.
Point 5: Line 63: a space before “The results…”
Response 5: We have made correction according to the reviewer’s comments.
Point 6: Line 64: homology or identity
Response 6: We have changed homology to identity.
Point 7: I still do not understand figure 6D and E
Response 7: We are very sorry, there are no figure 6D and E in the article. Are you talking about D E in figure 3? For Figure 3 D、E: Our process was roughly: We use two types of Arabidopsis, one was wild-type Arabidopsis, and the other was transgenic Arabidopsis, and extracted their total protein and passed through the column separately, respectively. Figure 3D and E were the results of SDS-PASE after column.
Because the genetic transformation system of Tamarix has not been established, it is very difficult to study interaction proteins with Thtrx5 in Tamarix. At the same time, through bioinformatics analysis, the identity of this gene and AtTRX-M4 of Arabidopsis reached 94.2%, and the conservative domain sequence of M subclass is identical. Therefore, it is feasible to analyze the interaction proteins with thtrx5 under salt stress with Arabidopsis as the material.
Point 8: Figure 4: is it WT Arabidopsis? How can you check the expression of Tamaris gene in Arabidopsis?
Response 8: We apologize for the lack of clarity in the previous statement. WT refers to wild type of Tamarix hispida. Studies have shown that homologous genes have similar expression patterns in different species, See reference 14. ThTrx5 has a high identity with AtTRX-M4 of Arabidopsis belonging to m class, up to 94.2%, and their conservative domain sequence of M subclass is identical. Therefore, we chose 3h as the transcriptome and pull-down time. We re-added in the article. See lines 152-154 for details, materials and methods related to experiments are added in sections 4.1.1, 4.1.2 and 4.2.3.
Point 9: I still do not understand figure 6D and E
Response 9: the same as Response 7.
Point 10: Figure 4: is it WT Arabidopsis? How can you check the expression of Tamaris gene in Arabidopsis?
Response 10: the same as Response 8.
This manuscript is a resubmission of an earlier submission. The following is a list of the peer review reports and author responses from that submission.
Round 1
Reviewer 1 Report
This manuscript reports the generation and analysis of Arabidopsis thaliana transgenic lines overexpressing the ThTrx5 gene from Tamarix hispida (a salt-resistant woody shrub) encoding a plastid thioredoxin protein of the m subclass of family I.
The authors studied salt tolerance in these overexpression (OE) lines by measuring the fresh weight and glutathione content of two of them in response to 150 mM NaCl. They also carried out pull-down assays and transcriptome analyses in order to determine the molecular mechanisms of the halotolerance of the OE lines. The authors conclude that ThTrx5 OE improves Arabidopsis salt tolerance likely through ThTrx5 interaction with CAT3, BAS1, PPI, GAPDH and ATP synthase proteins, regulating biological and metabolic pathways involving genes participating in redox processes, hormone signaling pathways and transcription factors.
The level of language is good enough for me and this work can be interesting for the readers of the IJMS. Notwithstanding, in the opinion of this reviewer, there are several shortcomings and serious flaws that prevent recommendation for acceptance in the present form.
Major concerns
The physiological and phenotypic characterization of the salt tolerance of the OE lines is incomplete and rather superficial. Only fresh weight of seedlings 7 d post-germination and after 48 h of treatment with 150 mM NaCl was determined and differences between WT and OE lines were rather small, and apparently not statistically significant between OE2 and WT (lines 83-85: “fresh weights of OE1 and OE2 plants were 1.3-fold and 1.2-fold higher than that of WT plants, respectively; Figure 2B)”. In addition, in lines 81-83 it can be read: “Transgenic and non-transgenic (WT) lines at 7 d post-germination were treated with 150 mM NaCl for 48 h then were phenotypically compared, revealing that leaves of WT plants were obviously chlorotic and yellow, while most plants of OE1/OE2 lines grew normally (Figure 2A)”. However, Fig. 2A only shows seedlings displaying cotyledons and not leaves and seedlings are too small to clearly see these differences between OE lines and WT.Therefore, I strongly recommend to study other body parameters in response to salt stress, such as the main root length, and also take better pictures of the plants clearly showing the differential effects of salt on WT and OE lines. In line with this, main roots of OE lines do not appear to be longer than those of WT plants (Figure 2A). Germination and seedling establishment in the OE lines in response to salt should also be determined, in order to know their early response to salt stress.
I also have some additional questions:
- The authors do not explain why they choose 3 h of exposure to 150 mM NaCl to analyze ThTrx5 putative regulatory mechanisms during salt stress. Why not a longer exposure and/or a higher concentration of NaCl?
- GSSG content of transgenic and WT control lines was determined at 0 h and 24 h of stress treatment. Why not at 48 h after exposure to salt stress, when fresh weight was measured?
- Was the fresh weight of WT and OE lines determined in the absence of stress?
Pull Down section requires clarification. In lines 108-111 it can be read: “After binding of the fusion protein to a nickel column, total protein of transgenic or WT A. thaliana plants was applied to the column and proteins that specifically bound to column-bound ThTrx fusion protein were retained after washing, then were eluted and detected as differential bands by SDS-PAGE (Figure 3A) followed by Western blotting (Figure 3B, 3C)”. I wonder if the transgenic lines and WT plants were exposed to salt stress prior to the protein extraction.Figure 3 legend. The antibodies used in the Western blot should be indicated.
It is not clear to me what are the differences between Fig. 3A, B and C.
The procedure followed in the mass spectrometry analysis should be indicated in Materials and methods.
Figure 4 is missing! Confirmation of the RNA-seq results requires to perform qRT-PCR experiments. This must be done to validate the expression results and conclusions of this section. Numbers of differentially expressed genes (DEG) in section 2.3.2 and 2.3.3 are different. In section 2.3.2, 500 and 194 DEG are reported between OE lines and WT plants at 0 h and 3 h of NaCl stress, respectively. However, in section 2.3.3, the number of DEG in the WT 0 h vs OE lines at 0 h are more than 5.000 and in the WT 3 h vs OE 3 h nearly 2.000. In my opinion, the Discussion is too speculative, taking into account the reported results. A little more concreteness would be appreciated.Minor concerns
The statistical method used in Figure 2 should be indicated. I suggest to move Figure 1 to Supplemental material. It is unnecessary to show all these gels in the manuscript. line 156. “cell component” should be “cellular component”. The in silico tool used to perform the analysis of GO enrichment of DEGs is not indicated. The most significantly enriched GO terms in the OE lines vs. the WT in response to salt stress should be commented in section 2.3.3 Sections 2.3.4-2.3.6. Full names for all the DEGs should be shown (see for instance ERF53 (AT2G20880.1), GATL2 (AT3G50760.1), SQS2 (AT4G34650.1) and GRP5 (AT3G20470.1) in page 7 or FSD1 (AT4G25100.2), PRX9 (AT1G44970.1), FER1 (AT5G01600.1) and FER4 (AT2G40300.1), among others, in page 8). Unify nomenclature. For instance, Arabidopsis thaliana, A. thaliana and Arabidopsis are interchangeably used throughout the document. line 91. ThTrx5 should be in italics. lines 100, 101 and 102. Arabidopsis thaliana should be in italics. line 122. ThTrx5 should be in italics.Reviewer 2 Report
In this paper, Luan et al. cloned ThTrx5 from Tamarix hispida, a salt-resistant woody shrub.
cloned ThTrx5 -a thioredoxin of the m class- from Tamarix hispida, a salt-resistant woody shrub. They generated 5 ThTrx5-overexpressing transgenic Arabidopsis thaliana lines. They used two of them in this study.
Using RNA-seq, they identify genes whose expression level is different in overexpressing lines versus WT, in control- and salt-treated- conditions. From the Differentially expressed genes, they then focus on transcription factors, genes related to hormone signaling and redox.
Using pull-down techniques, they show that in Arabidopsis thaliana, ThTRX5 does interact with a catalase 3, a glyceraldehyde-3-phosphate dehydrogenase, a peptidyl-prolyl cis-trans isomerase (CYP20-3), a ATP synthase subunit beta and a 2-Cys peroxiredoxin BAS1-like protein. They do nothing more of these results. Are these proteins chloroplastic, as ThTRX5 is supposedly plastidial (being of the m class)? It is not discussed. Anyway, where is TrTRX5 when overexpressed in A. thaliana?
Line 15: what do you mean by “ThTrx5 transcriptional expression »? transcriptome in ThTrx5-overexpressing transgenic Arabidopsis thaliana lines? Or the the expression of ThTRXh5 was verified? In the sentence “ThTrx5 transcriptional expression » goes with « RNA-seq » so please be clearer.
I would delete the sentence. And only precise “by RNA-seq » after “differential gene expression”.
What do you mean by “interactions of ThTrx5 with differentially expressed proteins ». It is the interactions with other proteins that you look at, not specifically with “differentially expressed proteins”.
Line 50: why mention GR? It is quite confusing.
Figure 1 is very muck bench book. Where are the data showing the overexpression of ThTRX5 in Arabidopsis thaliana? It is only panel E that matter. I do not understand the interest of B, C and D. In the caption, (A) and (B) are written in bold while (C), (D) and (E) not.
Line 83, it would have been nice to quantify chlorophyll.
Line 111, 112: in the text you write that Figure 3A is SDS-PAGE and figures 3B and 3C are Western. But in figure 3 caption, A is Western blot, B a SDS PAGE and C SDS POAGE. It is very confusing, and not clear at all.
Line 113, why “initially”?
Line 119/120. What is “B: SDS-PAGE identification of wild-type Arabidopsis interaction protein SDS-PAGE identification of ThTrx5 interaction protein.”? “Interaction protein” is for “interacting proteins”?
Lines 121 and 123: what do you mean by “column”: column-bound?
Line 1478: I do not see figure 4.
Line 154: VS is not necessarily clear. Write vs. in italics, or, best, versus. Change through the text.
Figure 5. There is no legend in the y-axis. Cannot you mingle A and B, with a color code, so that we can make the difference?
Line 150 and after, you should in the text also describe the subclasses. If not, it has no sense. What is interesting is the subclasses inside Biological process, cellular component and molecular function. The goal of GO classification I not to compare Biological process vs cellular component and vs molecular function, but inside each of them compare the subclasses.
Lines 164/165: the sentence is meaningless.
Figure 6, write after the gene name the AGI so that reader understand that it is one gene per line/-. Idem for figures 7 and 8.
Line 192; how do you select the DFG involved in hormone signaling pathway. Where do you retrieve this list. Besides it is not clear which genes is related to which. Find a way to indicate
In conclusion, all the paper is based in overexpressing a Tamarix hispida TRX gene in Arabidopsis thaliana. The interest of such heterogenous expression is questionable, because in fact it more documents the role of TRX in A. thaliana than in T. hispida. However, it can be a brick in a story. The problem that the fact the overexpressing plants do better resist the salt is poorly documented. More parameters should be teste, like may be survival test after putting back in no salt condition, chlorophyll measurement etc.. Figure 3 is not clear, and the authors do nothing of it…. The GO classification is very badly described. What is interested is the subclasses, not comparing biological process versus Cellular component versus Molecular function. The hormone data is not clear. Please for each hormone use a set of genes typical of the responses and signaling and do either a bar diagram or a heatmap. For instance ,for SA please do a heatmap of the genes involved in SA response or signaling as defined in Janda and Ruelland review. What about PR1 expression. The assumption that “ThTrx5 Regulates Biological and Metabolic Pathways via Protein Interaction” is not proven. This protein regulates metabolic pathways, and it interacts with proteins. That is all that can be said.